# COMPOSITIONAL META-LEARNING THROUGH PROBABILISTIC TASK INFERENCE

## ABSTRACT

To solve a new task from minimal experience, it is essential to effectively reuse knowledge from previous tasks, a problem known as meta-learning. Compositional solutions, where common elements of computation are flexibly recombined into new configurations, are particularly well-suited for meta-learning. Here, we propose a compositional meta-learning model that explicitly represents tasks as structured combinations of reusable computations. We achieve this by learning a generative model that captures the underlying components and their statistics shared across a family of tasks. This approach transforms learning a new task into a probabilistic inference problem, which allows for finding solutions without parameter updates through highly constrained hypothesis testing. Our model successfully recovers ground truth components and statistics in rule learning and motor learning tasks. We then demonstrate its ability to quickly infer new solutions from just single examples. Together, our framework joins the expressivity of neural networks with the data-efficiency of probabilistic inference to achieve rapid compositional meta-learning.

## 1 INTRODUCTION

Learning a new task rarely requires starting from scratch. For example, to cook a new dish you can rely on previously learned techniques like chopping or frying, or to play a new piano sonata you can incorporate fragments of familiar movements. Having learned a set of related "training" tasks (or dishes or sonatas) facilitates the acquisition of the new "test" task, a process known as "learning-to-learn" or "meta-learning" (Harlow, 1949; Hospedales et al., 2020). But these examples highlight a particularly powerful way of meta-learning: both involve reusing components from previous tasks in new configurations (Lake et al., 2015; 2017). This is a form of compositionality, which affords combinatorial generalisation by reusing a small number of elements in endless new combinations (Fodor, 1975; Battaglia et al., 2018). Here, we will propose a model that exploits compositionality to discover common components across solutions and learn how to combine those components to rapidly solve new tasks.

Such reusable components of computation emerge spontaneously in recurrent neural network (RNN) models trained to perform many tasks, a setting called "multi-task learning" (Caruana, 1997; Yu et al., 2020). For example, in an RNN trained on a suite of rule learning tasks, the same subnetwork activates across tasks that require evidence integration, and another subnetwork when producing a response (Yang et al., 2019). These components are implemented as "dynamical motifs", recurrent dynamics that carry out specific computations and are shared across tasks that require those computations (Driscoll et al., 2024). Importantly, these motifs support meta-learning: a new task that relies on a previously learned motif is acquired much faster than a task that requires a new motif. Our model builds on such dynamical motifs through a set of learnable modules that each implement an isolated but reusable computation.

To solve the task at hand, our model needs to combine these modules. But these combinations are often richly structured (Sutton et al., 1999; Markowitz et al., 2023). For example, the evidence integration module may usually be followed by the response module, but never the other way around. We therefore introduce a separate learnable model, which we call the gating network, that selects which module is activated when. This model acts like the gating function in a mixture-of-experts setup (Jacobs et al., 1991), or like the routing function in modular networks (Rosenbaum et al.,

2017). Importantly, this separation between modules and gating provides a strong inductive bias for the gating network to focus on the statistics of module combinations, rather than on the specifics of module dynamics. This architecture therefore encourages the gating network to learn the grammar that generates tasks, while the modules learn the tasks' syllables.

Together, the gating network and the modules thus learn a generative model of the "train" tasks. By casting our framework explicitly as a probabilistic generative model, we can solve new "test" tasks through probabilistic inference (Nguyen et al., 2021). That is fundamentally different from common meta-learning approaches (Finn et al., 2017; Li et al., 2018; Nichol et al., 2018; Rosenbaum et al., 2017; Ponti et al., 2022; Chitnis et al., 2019) which, through architecture or training, aim to minimise the parameter updates required to learn test tasks. By inferring rather than learning new solutions, we avoid parameter updates altogether. Instead, we compose solutions from modules, constrained by their statistics, guided by task feedback.

We show that we can simultaneously learn the gating network and modules by recovering ground truth modules and statistics in two abstract domains. Then, we demonstrate an inference procedure on our learned generative model that can infer new tasks from minimal experience, even when task feedback is sparse. Together, these results provide a framework for rapid acquisition of new tasks through compositional meta-learning.

## 2 RESULTS

### 2.1 A GENERATIVE MODEL OF TASKS

Our goal is to rapidly find a solution to a new (or test) task $q^{(Q+1)}$, after having encountered a set of previous (or training) tasks $\{q^{(i)}\}_{i=1...Q}$. We define a task $q^{(i)}$ as a data-generating process $q^{(i)} = \{f_t^{(i)}\}_{t=1...T}$ of $T$ timesteps, where $f_t^{(i)}$ are arbitrary functions that we need to learn. Given a timeseries of inputs $\{\mathbf{x}_t\}_{t=1...T}$ (for example, i.i.d. sampled from a Gaussian), the task thus generates output timeseries $\{\mathbf{y}_t = f_t^{(i)}(\mathbf{x}_{1:t})\}_{t=1...T}$. We refer to a timeseries of $\{(\mathbf{x}_t, \mathbf{y}_t)\}_{t=1...T}$ input-output pairs for a particular task as an episode. Our model needs to discover the data-generating process $\{f_t^{(Q+1)}\}_{t=1...T}$ to produce the correct output $\{\mathbf{y}_t\}_{t=1...T}$ given input $\{\mathbf{x}_t\}_{t=1...T}$ for the test task $q^{(Q+1)}$, from a minimal number of episodes. To achieve that, it needs to learn the commonalities across the training tasks $\{q^{(i)}\}_{i=1...Q}$, and figure out how to apply those in the test task. Importantly, we assume that many real-world tasks are modular: they generate data through varying combinations of sub-processes. In other words, there is a limited number of modules $f_t^{(i)}$ that get reused in different orders across tasks. Our model therefore needs to learn two key characteristics of the training tasks. First, it must isolate the modules to learn within-module dynamics; and second, it must extract how they are combined to learn between-module dynamics.

Our model separates these two learning objectives through its architecture (Figure 1a; Appendix A.1). The between-module dynamics are captured by a gating network, which selects from a set of module networks that each capture different within-module dynamics. We implement this through a gating RNN $G_{\boldsymbol{\theta}}$ that decides at each timestep which module $\mathbf{z}_t$ is activated from a set of module RNNs $\{M_{\boldsymbol{\phi}}^z\}_{z=1...N}$

$$\boldsymbol{g}_t = G_{\boldsymbol{\theta}}(\mathbf{x}_t, \boldsymbol{g}_{t-1}, \mathbf{z}_{t-1}) \tag{1}$$

$$\mathbf{z}_t \sim \mathrm{Cat}(\boldsymbol{W}_G \boldsymbol{g}_t) \tag{2}$$

where $\boldsymbol{W}_G$ is a linear projection from the hidden state $\boldsymbol{g}_t$ of $G_{\boldsymbol{\theta}}$ to the number of modules $N$ and Cat denotes a categorical distribution. The selected module gets to process the input to produce an output

$$\boldsymbol{m}_t = M_{\boldsymbol{\phi}}^{\mathbf{z}_t}(\mathbf{x}_t, \boldsymbol{m}_{t-1}) \tag{3}$$

$$\mathbf{y}_t \sim \mathrm{MVN}(\boldsymbol{W}_M \boldsymbol{m}_t, \sigma \boldsymbol{I}) \tag{4}$$

where $\boldsymbol{W}_M$ is a linear projection from the hidden state $\boldsymbol{m}_t$ of $M_{\boldsymbol{\phi}}^z$ to the output dimension $d_y$, $\sigma$ is a learnable standard deviation with $\boldsymbol{I}$ the $d_y$ dimensional identity matrix, and MVN denotes a multivariate normal distribution. For notational convenience, we'll group all learnable parameters as $\Lambda = \{\sigma, \boldsymbol{\theta}, \boldsymbol{\phi}, \boldsymbol{W}_G, \boldsymbol{W}_M\}$.

Crucially, we don't provide any of the networks with the task identity. Whereas indicating the current task is standard in brain-inspired multi-task learning (Yang et al., 2019; Duncker et al., 2020; Márton et al., 2022; Riveland & Pouget, 2024; Driscoll et al., 2024), our model learns from task output feedback alone. The resulting model therefore doesn't specify how to solve one particular task. Instead, it learns the underlying dynamics across all tasks, formalised as a probabilistic generative model in Equations 1-4 (Figure 1b). Once that generative model has been learned from the training tasks, finding a solution to the test task is a matter of inference rather than learning - there's no need for parameter updates. Concretely, having learned the within-module dynamics, solving the test task only requires finding the module sequence that best explains the test task data. And having learned the between-module dynamics, the space of possible sequences will be highly constrained. This process resembles learning and inference in a classic hidden markov model (HMM), but one where the transition matrix is replaced by the gating RNN and the emission matrix by the module RNNs. These RNNs greatly enhance expressivity: the gating RNN can learn long-distance non-Markovian dependencies, and the module RNNs can learn arbitrary emission functions. Meanwhile, we can still rely on the efficient inference machinery afforded by probabilistic models.

We apply these probabilistic inference methods both for inferring the solution for the test task and to compute the loss function used to train the RNNs. Specifically, we use particle filtering (Gordon et al., 1993; Doucet & Johansen, 2009) to obtain the posterior module selection of a test episode and to compute the marginal likelihood of the training episodes (although our model is agnostic to the choice of approximate inference method - others may also work). Briefly (Figure 1c; Appendix A.2), given a particle system $\{k_{t-1}\}_{k=1...K}$ where $k_{t-1} = \{z_{1:t-1}^{(k)}, g_{1:t-1}^{(k)}, m_{1:t-1}^{(k)}\}$ we sample a module transition through Equations 1-2 to get module selection $\{z_t^{(k)}\}_{k=1...K}$. By evaluating the sampled modules via Equations 3-4, we obtain output mean $\{\mu_t^{(k)}\}_{k=1...K}$ where $\mu_t = W_M m_t$, which provides the particle's likelihood of the current target output $y_t$:

$$l_t^{(k)} = p(y_t|z_t^{(k)}; \Lambda) = p(y_t|\mu_t^{(k)}; \Lambda) \tag{5}$$

We then sample particles for propagation to the next timestep from the (normalised) likelihoods

$$k_t \sim \text{Cat}(l_t^{(k)} / \sum_{i=1}^{K} l_t^{(i)}) \tag{6}$$

via stratified resampling (Appendix A.2). This particle system provides us with two important quantities. The distribution of modules selected by resampled particles from Equation 6 approximates the posterior $p(z_t|y_{1:t})$. We need that to infer the best sequence of modules for the test task. The particle likelihood before resampling in Equations 5 determines the loss for learning the model parameters on the training tasks. We need that to calculate the marginal likelihood at each timestep

$$p(y_t|y_{1:t-1}; \Lambda) = \frac{1}{K} \sum_{i=1}^{K} l_t^{(i)} \tag{7}$$

so that we can get the marginal likelihood across the whole timeseries

$$L = p(y_{1:T}; \Lambda) = \prod_{t=1}^{T} \frac{1}{K} \sum_{i=1}^{K} l_t^{(i)} \tag{8}$$

We optimise model parameters $\Lambda = \{\sigma, \theta, \phi, W_G, W_M\}$ through gradient descent on negative log marginal likelihood $-\log(L)$, backpropagating the loss through the particle filter on the training tasks (Appendix A.2). We use the gumbel-softmax reparameterisation trick to calculate gradients through the sample of Equation 2.

In summary, we learn a probabilistic generative model that separates within-module dynamics (acting as 'task syllables') from between-module dynamics (to form a 'task grammar'). We train this model by maximising the marginal likelihood of the training tasks through backpropagation. Then we find the best module sequence to solve the test task by probabilistic inference. Importantly, this means that test tasks are solved without any parameter updates. In the remainder of this paper, we will demonstrate the power of this approach on rule learning and motor learning tasks.

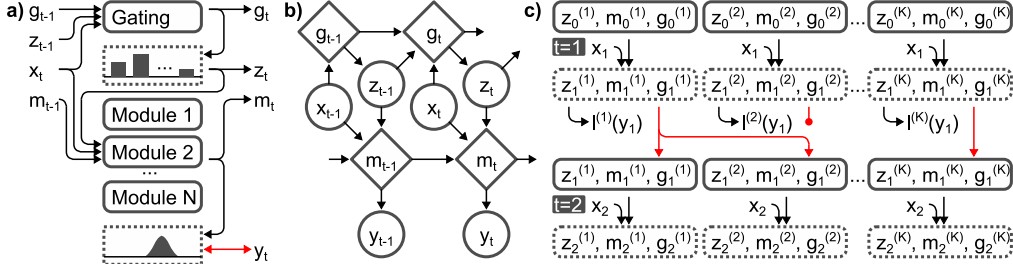

Figure 1: Model overview. **a)** Model architecture. The model consists of a gating RNN that for a given gating hidden state $g_{t-1}$, previously activated module $z_{t-1}$, and input $\mathbf{x}_t$, parameterises a discrete probability distribution from which the currently active module RNN $z_t$ is sampled. The selected module RNN processes the input $\mathbf{x}_t$ and module hidden state $\mathbf{m}_{t-1}$ to define an output distribution for $\mathbf{y}_t$. **b)** Graphical model. The model learns a probabilistic generative process with stochastic variables (circles) $z_t$ and $\mathbf{y}_t$ that depend on input $\mathbf{x}_t$ and the deterministic model hidden states (diamonds) $g_t$ and $\mathbf{m}_t$. Conceptually, this expands a HMM by replacing the transition and emission matrices by input-dependent RNNs. **c)** Particle filter schematic. To perform inference in this generative model, we define a particle system of $K$ particles (top row). At a given timestep, we sample module activations to calculate likelihoods $l^{(i)}$ of data $\mathbf{y}_t$ for each particle (second row). We resample particles from these (normalised) likelihoods (red arrows: particle (1) is sampled twice, whereas particle (2) is terminated) to reflect the module posterior $p(z_t|\mathbf{y}_{1:t})$ (third row), and continue this process for the next timestep (bottom row).

## 2.2 RECOVERY IN ABSTRACT RULE LEARNING

We first apply our model to an abstract rule learning task to show it can recover ground truth within-module and across-module dynamics. This task mimics the hardest variants of context-based rule switching because the model needs to simultaneously learn what the rules are and when to apply them, from uninformative input. We arbitrarily use vector dimension shifts as rules, with stay-switch transition dynamics, and random 6D vectors as inputs. There is nothing special about these choices; what matters is that they provide ground truth within-module and across-module dynamics that we can probe in the model. The data-generating process is given by

$$\mathbf{y}_t = S_s(\mathbf{y}_{t-1}) + \mathbf{x}_t \tag{9}$$

where $\mathbf{y}_t, \mathbf{x}_t \in \mathbb{R}^6$, $\mathbf{x}_t \sim \mathcal{N}(0, \mathbf{I}_6)$, $\mathbf{y}_0 = 0$, and $\{S_s\}_{s=0\ldots5}$ is the set of 6D shift operations that shift the value of vector entry $i$ to entry $\mathrm{mod}(i + s, 6)$. A task is defined as a sequence of shift operations that determines the input-output mapping at each timestep. Importantly, these sequences have a very specific structure. Each task consists of three shift operations, and each shift operation repeats a fixed number of timesteps: 3 timesteps for $S_0$ and $S_1$, 4 timesteps for $S_2$ and $S_3$, and 5 timesteps for $S_4$ and $S_5$. Given this structure, an example task may sequence $S_1$ then $S_4$ then $S_2$ so that $s$ takes on the values 1,1,1,4,4,4,4,2,2,2,2 over timesteps. Thus, from episodes generated by the training tasks, the module RNNs must each learn one of the shift operations, and the gating RNN needs to learn the sequence regularities.

Indeed, we find that our model successfully learns the shift operations and their sequence statistics. As the total negative log marginal likelihood over training decreases, the accuracy of the module RNNs and the gating RNN rises to plateau at 1 (Figure 2a). This means that when we provide each module RNN after training with a set of one-hot $\mathbf{y}_{t-1}$ and all-zero $\mathbf{x}_t$ probe vectors, they produce outputs $\mathbf{y}_t$ with shifted entries exactly like the ground truth shift operations (Figure 2b; modules are post-hoc reordered to match the order of shift operations). Moreover, when we provide the gating RNN after training with a history of module selections, the output probability distribution across modules reflects the ground truth regularities (Figure 2c). The learned transition matrices highlight that after one or two repetitions of a module, that same module should be selected. After that, the gating network switches to any other module - but only for $S_0$ and $S_1$, which are each repeated 3 times. In the next step, $S_2$ and $S_3$ switch, and in the next step, $S_4$ and $S_5$ switch. Importantly, these immediate changes in the transition matrix depending on the history of module selection indicate that the gating RNN has learned the underlying strongly non-Markovian statistics; a HMM would

not be able to capture those. We have so far assumed an equal number of task operations and model modules, but we still find correspondence in learned modules and transitions when there's a data-model mismatch (Figure A1). If there are more modules than operations, the redundant modules remain unused (Figure A1a,b); if there are fewer modules than operations, the modules approximate a subset of the operations (Figure A1c,d).

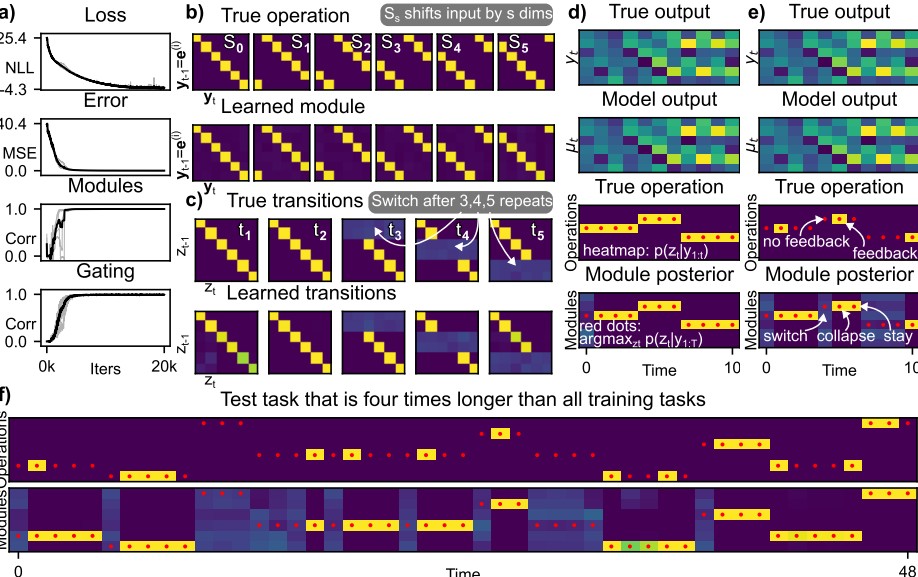

Figure 2: Rule learning. **a)** Learning curves. The training loss (negative log marginal likelihood) and task performance (mean squared error) decrease while the module and gating accuracy (correlation with ground truth operations and transitions) plateaus at 1 (grey lines: five individual seeds; black line: mean across seeds). **b)** Learned operations. Each of the six (columns) true shift operations $S_s$ (top row) shift their input entries by $s$ dimensions. This is shown by plotting in each matrix row $i$ the outcome $\mathbf{y}_t$ for unit vector input $\mathbf{y}_{t-1} = \boldsymbol{e}^{(i)}$, so that matrix column $j$ indicates where the 1 in the input ends up in $\mathbf{y}_t$. The six (columns) learned modules (bottom row) perform the same operation. **c)** Learned transitions. The true tasks incorporate a specific structure to operation sequences, shown by history-dependent transition matrices (top row). Each matrix row $i$ indicates the probability of the next module activation given that the previously selected module was module $i$ (first column), that the previous two selected modules were module $i$ (second column), previous three selected modules were module $i$ (third column), *et cetera*. The learned transitions (bottom row) reproduce the true transition pattern. **d)** Example test task. After training, the model infers a solution that accurately produces (second row) the desired output for a held-out task (top row). This test task's sequence of shift operations (third row; each matrix row represents one shift operation and red dots indicate the true underlying operation sequence) is accurately recovered by the model's module posterior (bottom row; matrix rows are module selection probabilities so that each column plots $p(\mathbf{z}_t|\mathbf{y}_{1:t})$ with red dots showing the maximum a posteriori sequence $\operatorname{argmax}_{\mathbf{z}_t} p(\mathbf{z}_t|\mathbf{y}_{1:T})$. **e)** Sparse feedback example. The model still infers an accurate test task solution when feedback is provided in a small minority of timesteps (marked in yellow in the third row). **f)** Extended task example. Even on a test task that is four times longer than the training tasks, the model infers the correct solution, despite sparse feedback.

## 2.3 ONE-SHOT TASK ACQUISITION

With this learned generative model available, we can now infer solutions to new test tasks. We plot the posterior module selection given the data so far $p(\mathbf{z}_t|\mathbf{y}_{1:t})$ at each timestep (Figure 2d, bottom row, heatmap), and the model output (Figure 2d, second row) for the maximum a posteriori sequence of modules $\operatorname{argmax}_{\mathbf{z}_t} p(\mathbf{z}_t|\mathbf{y}_{1:T})$ (Figure 2d, bottom row, red dots). These outputs show that the model reaches the correct solution, without parameter updates, based on a single data episode of the test task. But an even stronger demonstration of the power of probabilistic inference can be

made under sparse feedback conditions (Figure 2e). Sparse feedback means that $\mathbf{y}_t$ is available only at intermittent timesteps (Figure 2e, third row, yellow timesteps). At timesteps without feedback, we cannot rely on the target likelihood to infer the appropriate module. Instead, we need to track different hypotheses on all the possible sequences of module selections until new feedback arrives. Here the gating RNN proves particularly effective: it constrains which module sequences are possible in the absence of feedback. This is reflected by the posterior $p(\mathbf{z}_t|\mathbf{y}_{1:t})$ in episodes with sparse feedback (Figure 2e, bottom row, heatmap). On a timestep with feedback, the posterior collapses for the subsequent timesteps without feedback - but only until the module has been repeated for the learned number of timesteps. After that, the posterior turns uniform across modules. This change is a product of the learned dynamics in the gating RNN (Figure 2c) because it is not signalled by $\mathbf{x}_t$ or $\mathbf{y}_t$. As soon as feedback returns, the posterior continues only from the hypothesis confirmed by this feedback. As a result, $\mathrm{argmax}_{\mathbf{z}_t} p(\mathbf{z}_t|\mathbf{y}_{1:T})$ traces the true module sequence at the end of the episode (Figure 2e, bottom row, red dots). The model thus overcomes feedback sparsity through constrained hypothesis testing. This remains effective far beyond what the model was exposed to during training: the model can solve test tasks that are four times as long as any of the training tasks (Figure 2f).

This capacity to infer a new test task from a single episode, even under sparse feedback, is specific to our architecture. We will compare it to three control models to support this claim. First, when we replace our model by a standard RNN that receives the same inputs, the RNN cannot solve the training or test tasks (Figure 3a). This is unsurprising as the network can't know what to do across tasks without task identity input. We therefore also train a standard RNN that receives additional task identity input (Figure 3b). This control model does learn to perform the training tasks, but performs poorly on the held-out test tasks, as the task identity of the test tasks will not have been present during training. It can be retrained on the test tasks, as we will investigate in the next paragraph, but this will require additional weight updates through gradient descent (and even more for sparse feedback). If instead we turn to our architecture, but replace the gating RNN by a uniform transition matrix, we are able to learn the training tasks and infer the test tasks (Figure 3c). However, when feedback is sparse, the model suffers from the lack of constraints on the transition structure and fails to do task inference. Only the full architecture successfully learns the training tasks and infers the test tasks even under sparse feedback (Figure 3d).

To demonstrate the effectiveness of one-shot test task inference, we compare it to the common alternative: learning to perform the test task by updating model parameters through gradient descent (Figure 3e). We use a RNN with task identity input as in Figure 3b that takes a gradient descent step after each test task episode. The learning curves show clear signs of reuse: the model that learns the test task from scratch (blue) learns slower than any of the other models that have been pre-trained to perform the training tasks first. However, whether we pre-trained with standard gradient descent (orange) or model-agnostic meta-learning (Finn et al., 2017) (red) or meta-learning for domain generalisation (Li et al., 2018) (purple) doesn't make a difference. All learn on a timescale of hundreds of episodes which is qualitatively different from the single-episode inference (grey). The fact that freezing recurrent weights and relearning only the input weights of the pre-trained model (green) performs equally well indicates that the recurrent dynamics from pre-training are sufficient to solve the test task, and the network only needs to learn to map the test tasks onto these dynamics. This is no longer the case if the test task is longer than the training tasks (Figure 3f). Freezing the recurrent weights and retraining input weights (green) then doesn't reach the same asymptotic performance as (re)learning all weights (blue, orange). Because our model learns the general rules for sequencing modules, it automatically generalises to longer tasks without retraining (grey). On the other hand, if test tasks are incompatible with the learned modules or transition statistics, the episode likelihood will be a clear indicator of out-of-distribution data (Figure A1e) which could trigger retraining.

## 2.4 LEARNING AND INFERRING MOTOR SKILLS

Although usually considered separately from rule learning, motor learning is another domain where agents can greatly benefit from compositionality, by combining simple skills into complex behaviours (Ijspeert et al., 2013; Berg et al., 2023; Merel et al., 2019). Our framework naturally accommodates both. In a motor task where composite trajectories are generated from reusable chunks, the model isolates the chunks and infers how to recombine them into new trajectories. We define the skills in this task as sequences of translations, each step with an increased or decreased angle compared to the previous to create curvature, and an increased or decreased magnitude to create

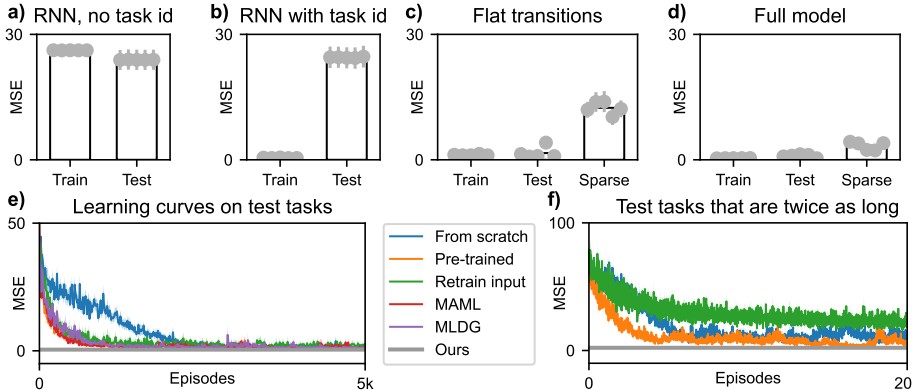

Figure 3: Control models. **a)** RNN control model. An RNN trained to perform the tasks with the same input as our model cannot learn either train or test tasks (grey dots: individual seeds, error bars s.e.m. across tasks; black bars: mean across seeds). **b)** RNN with task identity input. When the task identity is added to the inputs, the RNN performs well on the training tasks but cannot solve new test tasks without additional training. **c)** Task inference without gating network. As our model learns a generative model across tasks, it performs well on the training tasks and infers solutions to held-out test tasks from a single episode. However, without a gating network it performs poorly on tasks with sparse feedback. **d)** Full model. The full model learns training tasks and infers test tasks even if feedback is sparse. **e)** Test task learning curves, averaged across tasks. Training the parameters of an RNN with task identity input (as in b) through gradient descent on one test task episode at a time is faster after pre-training on training tasks (blue: no pre-training; orange: pre-train on training tasks by standard gradient descent; green: pre-train on training tasks, freeze recurrent weights, only retrain input weights; red: pre-train through MAML (Finn et al., 2017); purple: pre-train through MLDG (Li et al., 2018)), but qualitatively slower than single-episode task inference (grey). **f)** Test task learning curves on double length test tasks. When the test tasks are twice as long as the pre-training training tasks, the asymptotic performance for retraining with frozen recurrent weights (green) suffers.

acceleration or deceleration. Importantly, the skills have different durations: 3 steps for skill 0 and 1, 4 steps for skill 2 and 3, and 5 steps for skill 4 and 5. Tasks are then generated by concatenating three skills into a combined trajectory (Figure 4a). The module RNNs should therefore each learn to output the sequence of translations for a skill, and the gating RNN must learn when to switch between skills.

Again, the model learns accurate modules and their transition statistics from the training tasks. We plot the learned skills by running each module RNN in isolation for multiple timesteps and find that the learned translations closely match the true skills (Figure 4b). The module transitions learned by the gating RNN reflect the durations of the skills (Figure 4c). These results illustrate how the same principles for learning a generative model can be applied to rule and motor learning tasks. Nevertheless, there are also important differences between the two. First, the motor task here does not require input $\mathbf{x}_t$, as the output is just a sequence of translations independent of inputs. Second, in the motor task each module needs to track progress within the skill, because that matters for its output. To accommodate for these differences, we make two practical changes to the model for motor learning: we remove the input $\mathbf{x}_t$ and we reset the module hidden state $\boldsymbol{m}_t$ after a module switch. Additionally, we increase each module's expressivity through module-specific $\boldsymbol{W}_M^z$ and improve the efficiency of the particle filter transition proposals by sampling them from $p(\mathbf{z}_t|\mathbf{z}_{t-1})p(\mathbf{y}_t|\mathbf{z}_t)$ instead of $p(\mathbf{z}_t|\mathbf{z}_{t-1})$ during training (Appendix A.2).

This learned generative model of motor tasks allows us to rapidly acquire new test tasks. We plot the true skill sequence with the module posterior $p(\mathbf{z}_t|\mathbf{y}_{1:t})$ as well as the maximum a posteriori sequence $\text{argmax}_{\mathbf{z}_t} p(\mathbf{z}_t|\mathbf{y}_{1:T})$ to demonstrate successful task inference (Figure 4d, bottom). The motor task has the advantage that the model outputs are easily visualised: we show the true task trajectory as a thick line coloured by skill, the maximum a posteriori sequence as a solid line coloured by module (which overlaps exactly with the true trajectory, so we give it a white border), and the

pre-feedback hypotheses $p(z_t|z_{t-1})$ as dotted lines coloured by module (Figure 4d, top). These hypotheses are most illustrative in the sparse-feedback case (Figure 4e). As before, we find that the posterior $p(z_t|\mathbf{y}_{1:t})$ in the absence of feedback develops through time respecting the learned module durations. But now we can also plot the content of the parallel hypotheses during these periods. As indicated by the dotted lines in Figure 4e, once a module has been confirmed by feedback (grey circle), this module is continued until completion, after which different hypotheses branch out to each track the possibility of the next potential skill. As soon as feedback returns, only the confirmed hypothesis continues, and all the other branches are cut short.

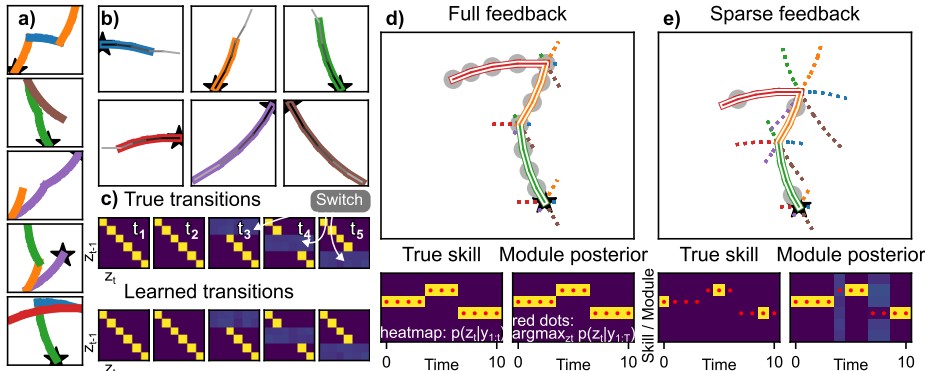

Figure 4: Motor learning. **a)** Example motor tasks. Each task consists of a sequence of three motor skills, shown as chunks of different colours, starting from the star. **b)** Learned skills. After training, each module (output of one module in thin grey lines from dark to light in each subplot) learns to perform a skill (thick coloured lines; one true skill in each subplot). **c)** Learned transitions. The history-dependent transition matrices, analogous to Figure 4c, show that the gating RNN learns to switch between skills depending on their true duration. **d)** Example test task. The model (thin solid line coloured by module, with white borders, and pre-feedback hypotheses in dotted lines) infers the new task trajectory (thick solid line, starting from star) from feedback at each step (grey circles) by selecting learned modules (bottom right, as in Figure 4d) that execute the true skills (bottom left). **e)** Sparse feedback example. When feedback is sparse (at grey circle locations, and timesteps marked in yellow at bottom left) the model tests module hypotheses (dotted lines) that branch out at skill switch points until feedback confirms the current skill. This is reflected by the posterior $p(z_t|\mathbf{y}_{1:t})$ collapsing to a single module at feedback timesteps (bottom right, heatmap) and an accurate maximum a posteriori sequence $\text{argmax}_{z_t} p(z_t|\mathbf{y}_{1:T})$ (bottom right, red dots).

## 3 DISCUSSION

Rapidly solving new tasks relies on thinking as well as learning, but meta-learning research has often focused on the latter. Here, we propose a framework that incorporates the former. We take a compositional approach by first learning the common components and the statistics of their combinations across training tasks. We then solve new tasks from minimal experience by thinking, or more specifically, compositional inference: reasoning about potential component configurations and testing these hypotheses against the current observations. Our results provide three current contributions. First, we formalise compositional meta-learning as inference in a learned probabilistic generative model with recurrent module and gating functions. Second, we define a neural network architecture for this model and specify a procedure for training its parameters and running inference. Third, we demonstrate how our architecture solves new tasks from single examples in rule learning and motor learning domains, even when feedback is sparse, without any parameter updates.

This is in stark contrast to traditional meta-learning approaches that explicitly focus on weight updates. Finn et al. (2017); Li et al. (2018); Nichol et al. (2018) adjust the training loop to learn parameters that require minimal gradient-based adaptation on new tasks. But exactly because they rely on parameter updates, they meta-learn qualitatively slower than our single-episode task inference. This inference is afforded by a modular architecture (Pfeiffer et al., 2024), which has been linked to compositional generalisation (Chang et al., 2018) as well as resource efficiency (Shazeer

et al., 2017; Fedus et al., 2022). In fact, modular computation emerges spontaneously in multi-task settings (Yang et al., 2019; Driscoll et al., 2024) (although optimal modularity may not (Csordás et al., 2021; Mittal et al., 2022)). Indeed, modular approaches display impressive generalisation (Goyal et al., 2020; Andreas et al., 2017) and meta-learning capacity (Márton et al., 2022; Duncker et al., 2020; Rosenbaum et al., 2017; Ponti et al., 2022; Chitnis et al., 2019; Hurtado et al., 2021). But these examples of modular meta-learning still require parameter updates on test tasks. The approach in Alet et al. (2019) doesn't, making it most similar in spirit to ours. It fixes module parameters after training, then searches the module configuration of test tasks through simulated annealing. We effectively replace this search by probabilistic inference on learned structure, greatly improving sample efficiency.

Solving new tasks without parameter updates is also related to in-activity (Hochreiter et al., 2001), in-memory (Santoro et al., 2016), and in-context (Brown et al., 2020; Oswald et al., 2023) learning, but those lack modularity and probabilistic reasoning - both key for compositionality in our framework. Indeed, the approach in Hummos et al. (2024) relies on compositional inference to achieve meta-learning without parameter updates, making it particularly closely related to ours. It proposes an efficient gradient update on a latent task embedding to rapidly discover new embeddings for test tasks. This allows for combining learned computations into new solutions by optimising activity (instead of weights as in Driscoll et al. (2024) or the green line in Figure 3e), but not for sequencing modules along learned transition statistics. Importantly, Hummos (2022) connects these latent embedding vectors to neural processing in the brain, by interpreting them as mediodorsal thalamic neurons that multiplicatively gate prefrontal cortex. This thalamic gating mechanism also offers a potential (biological) neural substrate for our sequential activation of modules, as it could gate different clusters of recurrent neurons in different temporal contexts (Zheng et al., 2024). Indeed, thalamic gating can be used to flexibly sequence cortical motifs to generate motor outputs (Logiaco et al., 2021). Together with the finding that motor learning is well-explained by probabilistic contextual inference (Heald et al., 2021), these studies draw intriguing parallels between our model and brain computation.

The results reported here serve as a proof-of-principle, so there are many ways to expand on them. Importantly, the number of modules is currently predefined and fixed. As a promising direction for future work, this could be addressed by applying the model in a (class-incremental (van de Ven & Tolias, 2019)) continual learning setting (Parisi et al., 2019; Ostapenko et al., 2021). In such a setting the model could dynamically add new modules if inference using the existing modules fails, signalled by low episode likelihood (Figure A1e). Solving new tasks through inference is particularly advantageous in continual learning because it circumvents catastrophic forgetting (McCloskey & Cohen, 1989), as no parameter updates are needed. Moreover, continual learning naturally affords training task curricula (Elio & Anderson, 1984). The current simultaneous learning of modules and gating risks training instability and local minima due to a 'chicken-and-egg' problem (Rosenbaum et al., 2019). The gating of modules is hard to learn if their functionality hasn't developed; the modules are hard to learn if the gating signal is inconsistent (Alet et al., 2019). Curriculum learning likely improves this situation (Chang et al., 2018; Lee et al., 2024).

The tasks that we designed to test our model are similarly proof-of-principle. Despite being synthetic and low-dimensional, these tasks are valuable because 1) they are hard: they require simultaneously learning arbitrary transformations and their sequencing from uninformative input, 2) they are controlled: we know the true modules and sequences so can verify what the model actually learns, 3) they demonstrate application in different domains: rule learning and motor learning. Moreover, the model's core ideas, namely slow synaptic learning of modules and their transitions across training tasks and fast inferential learning of new module combinations on test tasks, will apply to any problem with sequential modular structure. In addition to task domains, we note that our architecture also generalises to different gating and module components. Replacing the gating RNN by a modern transformer would allow for learning complex task grammars not unlike the one governing natural languages (Vaswani et al., 2017). Forcing the module RNNs to be low rank would implement a bias towards combinations of simple operations through low-dimensional dynamics (Mastrogiuseppe & Ostojic, 2018). Together, our results and suggestions chart a path towards the rapid composition of new solutions from learned elements across domains.

ETHICS STATEMENT

As this work doesn't involve human participants or real-world data, and introduces small-scale models only, we don't foresee any ethical ethical issues related to this paper.

REPRODUCIBILITY STATEMENT

To ensure reproducibility, we provide the full anonymised codebase (code for models, training, and analysis) as supplementary material. This includes scripts to generate every panel in the figures of this paper to directly reproduce all results reported here. We also provide the trained weights of all analysed models. We describe our choices of parameters in the appendices.

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

# A  APPENDIX

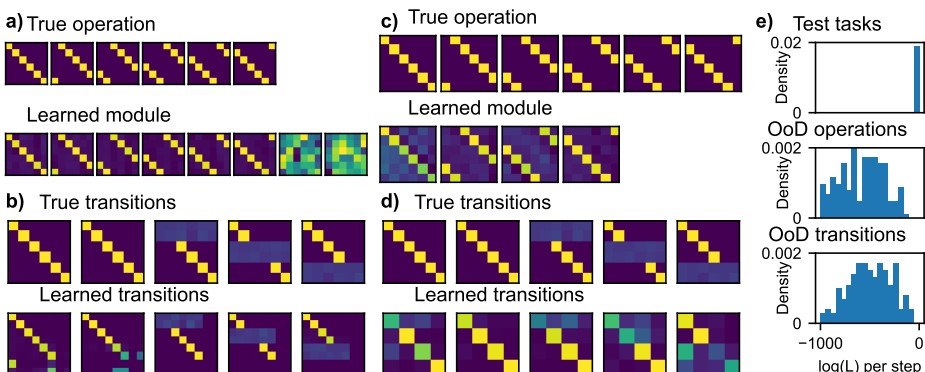

Figure A1: Data-model mismatch. Starting from the default rule learning task and model (Figure 2), we vary the model while keeping the same tasks (a-d) and vary the test tasks while keeping the same model (e). **a)** Shift operations (top) in the rule learning task with six operations, with learned modules (bottom) for a model with access to eight modules. These matrices contain in each row $i$ the output of an operation (top) or module (bottom) given the six-dimensional unit vector $i$ as input, as in Figure 2b. Six of the model's modules learn one ground truth shift operation each, the remaining two modules remain random. **b)** Ground truth (top) and learned (bottom) transitions for the eight-module model on the six-operation task. Each matrix plots transition probabilities for data operations (top) or model modules (bottom) given multiple repeats of the operation or module, as in Figure 2c. The transitions between the first six learned modules mirror the data operation transitions, whereas the remaining two modules remain unused. **c)** Shift operations (top) and learned modules (bottom) for a four-module model on the same six-operation task. The modules now learn (approximations of) a subset of operations, here operation $0, 2, 4, 5$, with an apparent preference for operations that repeat more timesteps $(2, 4, 5)$. **d)** Learned transitions (bottom) approximate the transition structure for these modules (top). **e)** Episode log likelihoods per timestep for in-distribution (top) and out-of-distribution (OoD; middle, bottom) test tasks. Here we use the default (six-module) model and training tasks as in Figure 2, but change the test tasks. If the test tasks are compiled from the same operations as the training tasks, through the same transition statistics, the likelihood of episodes from these held-out tasks is high. If the test tasks use different operations (here: random permutations) with the same transition statistics (middle), or the same operations but different transition statistics (here: random switches; bottom), the log likelihood of episodes is low.

## A.1  MODEL AND TRAINING

We implement all RNNs, meaning the gating network and each of the module networks, as simple Elman RNNs, each with a 32-dimensional single-layer hidden state. The control RNN in Figure 3a,b is a GRU (Cho et al., 2014) with $7 * 32 = 224$ hidden units to match the overall number in our model (but with many more weights, as there are no independent modules). The initial hidden state of these networks is a learned parameter vector. At each timestep, a linear projection from the gating hidden state parameterises a categorical probability distribution by assigning log probabilities to each category (Equation 2). We sample the selected module from these probabilities through the gumbel-softmax reparameterisation trick (Jang et al., 2017). That yields a soft (i.e. approximately but not strictly one-hot) activation vector across modules, which allows for gradient flow through the sampling step during training. After training, during inference on test tasks, we replace the activation vector by its hard (i.e. strictly one-hot in the argmax dimension) equivalent. To obtain the timestep's module hidden state, we sum the hidden state across all modules, weighted by the activation vector. A linear projection from the resulting module hidden state provides the output at the timestep (Equation 4).

Training the gating and module parameters simultaneously is prone to local minima and instability. To mitigate this, we found the initialisation of network weights to be important. Small initial weights, determined by weight factor $w_{init} = 0.01$ ($w_{init} = 0.001$ for motor learning), decrease

variability across seeds. Specifically, we initialise linear layer weights as Xavier uniform with gain $w_{init}$ and initialise linear layer bias (if present) as 0. For recurrent networks, we initialise input weights as Xavier uniform with gain $w_{init}$ and bias as 0; recurrent weights as orthogonal, then multiplied by $w_{init}$; and recurrent bias as 0. We then train our model's parameters by gradient descent on the negative marginal log likelihood (Equation 8) using the ADAM optimiser (Kingma & Ba, 2017) with learning rate 0.0003 (0.0001 for motor learning) for 20k iterations. We obtain this loss through particle filtering (Equations 5-8; also see Appendix A.2) with 250 particles. We use a batch size of 512, gradient clipping of 1.0, without regularisation or learning rate schedulers. For all models in the comparisons of Figure 3e,f and the mismatch experiments in Figure A1a-d we use learning rate 0.001, weight initialisation 1, batch size 128, and 5k training iterations on the training tasks.

## A.2 PROBABILISTIC INFERENCE

We use particle filtering to implement probabilistic inference in our learned generative model. Each particle carries the module hidden state $\boldsymbol{m}_t$ and the gating hidden state $\boldsymbol{g}_t$, as well as the selected module $z_t$. Tracking the RNN hidden states matters, because these contain long-range history dependencies; particles with the same $z_t$ may have different $\boldsymbol{m}_t$ and $\boldsymbol{g}_t$ because their module selection history differs. Additionally, we track each particle's ancestor. After updating the particle parameters following Equations 1-4, the ancestor indexes which particle the current particle was resampled from in Equation 6. That allows us to trace back the sequence of module activations with the highest likelihood at the end of the episode (Douc et al., 2009).

There are two technical details that majorly improve the efficiency of the particle resampling procedure described in Equations 1-6. The first is that we use stratified (also called balanced or systematic) resampling (Kitagawa, 1996; Doucet & Johansen, 2009) to implement Equation 6. Equation 6 defines a multinomial distribution that we repeatedly sample from to update the particle system with respect to the likelihood of the current observations. But rather than sampling from that distribution directly, we sample $u \sim \text{Uniform}(0, 1/K)$, and then follow a deterministic procedure for obtaining particles for the next step. We define a set of boundaries $b_k$ as the cumulative sum of the particle probabilities in Equation 6: $b_k = \sum_{i=1}^{k} l_t^{(k)} / \sum_{j=1}^{K} l_t^{(j)}$ and $b_0 = 0$. Then we select resampled particles by taking steps $b \in \{u + i/K\}_{i=0...(K-1)}$ and returning the particles for which $b_{k-1} \leq b < b_k$. This stratified sampling reduces degeneracy, where many particles carry identical information.

The second improvement to particle resampling efficiency only applies when training the model. That is because this method, referred to as guided particle filtering (Carpenter et al., 1999), relies on observations in the future. During inference, we assume our model doesn't have access to these future observations, so we stick to Equations 1-5 (known as bootstrap particle filtering (Doucet et al., 2001)). In guided particle filtering, we sample particle module selection $z_t \sim p(z_t|z_{t-1})p(\mathbf{y}_t|z_t)$ instead of $z_t \sim p(z_t|z_{t-1})$ as implemented by Equation 2. That means Equations 1-5 are replaced by

$$\boldsymbol{g}_t = G_{\boldsymbol{\theta}}(\mathbf{x}_t, \boldsymbol{g}_{t-1}, z_{t-1}) \tag{10}$$

$$\boldsymbol{m}_t^{z_t} = M_{\boldsymbol{\phi}}^{z_t}(\mathbf{x}_t, \boldsymbol{m}_{t-1}) \tag{11}$$

$$f_{z_t} = L_{\text{Cat}}(z_t; \boldsymbol{W}_G \boldsymbol{g}_t) L_{\text{MVN}}(\mathbf{y}_t; \boldsymbol{W}_M \boldsymbol{m}_t^{z_t}, \sigma \boldsymbol{I}) \tag{12}$$

$$z_t \sim \text{Cat}(f_z / \sum_{i=1}^{N} f_i) \tag{13}$$

$$l_t^{(k)} = \sum_{i=1}^{N} f_i^{(k)} \tag{14}$$

where $L_{\text{dist}}(x; \lambda)$ denotes the likelihood of $x$ under distribution $dist$ with parameters $\lambda$. We use guided particle filtering only for the motor learning task in the current work, but it is task-agnostic.

With this particle system in place, Equations 7-8 then specify the marginal likelihood that we use for training the model's parameters. Below, we provide additional details of how to arrive at these equations. They are the particle filter Monte Carlo estimate of the marginal likelihood, obtained by factorising $p(\mathbf{y}_{1:T})$ through time and replacing the integrals over latent variables by averages over particles. At time $t-1$ we have $K$ particles $k_{t-1} = \{z_{1:t-1}^{(k)}, \boldsymbol{g}_{1:t-1}^{(k)}, \boldsymbol{m}_{1:t-1}^{(k)}\}$ reflecting $K$ latent trajectories. Here $z_t$ indicates the selected module at time $t$, $\boldsymbol{g}_t$ the corresponding hidden state of the

gating RNN, and $\boldsymbol{m}_t$ the resulting hidden state of the module RNN. For each particle, we sample from the transition probability to obtain a predictive state $\{\mathbf{z}_t^{(k)}, \boldsymbol{g}_t^{(k)}, \boldsymbol{m}_t^{(k)}\}$. The likelihood of the observation $\mathbf{y}_t$ at time $t$ for this particle $k$ is given by the emission probability (Equation 5):

$$l_t^{(k)} = p(\mathbf{y}_t|\mathbf{z}_t^{(k)}; \Lambda) = p(\mathbf{y}_t|\boldsymbol{\mu}_t^{(k)}; \Lambda)$$

where $\boldsymbol{\mu}_t^{(k)} = \boldsymbol{W}_M \boldsymbol{m}_t^{(k)}$ with $\boldsymbol{W}_M$ readout weights included in $\Lambda$. Ultimately at each timestep we want the predictive likelihood

$$p(\mathbf{y}_t|\mathbf{y}_{1:(t-1)}; \Lambda) = \int p(\mathbf{y}_t|k_t; \Lambda) p(k_t|\mathbf{y}_{1:(t-1)}; \Lambda) dk_t$$

The particle filter provides samples $k_t$ which approximately follow $p(k_t|\mathbf{y}_{1:(t-1)}; \Lambda)$ so that we get a Monte Carlo approximation of the integral:

$$p(\mathbf{y}_t|\mathbf{y}_{1:(t-1)}; \Lambda) \approx \frac{1}{K} \sum_{k=1}^{K} p(\mathbf{y}_t|k_t; \Lambda)$$

Therefore, using Equation 5, we get

$$p(\mathbf{y}_t|\mathbf{y}_{1:(t-1)}; \Lambda) \approx \frac{1}{K} \sum_{k=1}^{K} l_t^{(k)}$$

which is Equation 7. The marginal likelihood of the whole sequence factorises through time:

$$L = p(\mathbf{y}_{1:T}|\Lambda) = \prod_{t=1}^{T} \frac{1}{K} \sum_{k=1}^{K} p(\mathbf{y}_t|\mathbf{y}_{1:(t-1)}; \Lambda)$$

Substituting the particle estimate of each timestep's likelihood from Equation 7 yields

$$L = p(\mathbf{y}_{1:T}|\Lambda) \approx \prod_{t=1}^{T} \frac{1}{K} \sum_{k=1}^{K} l_t^{(k)}$$

to obtain Equation 8. In practise we calculate the log likelihood

$$\log(L) = \sum_{t=1}^{T} \log(\frac{1}{K} \sum_{k=1}^{K} l_t^{(k)})$$

and minimise $-\log(L)$ by backpropagation through the particle filter.

