# OpenReview forum: "Compositional meta-learning through probabilistic task inference"
_ICLR.cc/2026/Conference — Submitted to ICLR 2026_

### Official Review · Reviewer_Gbhf · 2025-10-27

**Soundness:** 2
**Presentation:** 3
**Contribution:** 2
**Rating:** 2
**Confidence:** 5

**Summary:**

The authors propose to recast meta-learning as a probabilistic inference problem. They develop a RNN model, learning specific “subtasks” or motifs of a master task, acting as a task-solving backbone. In parallel, they implement a gating network that allows to guide information processing in the backbone model and select the appropriate RNN module sequence to solve a given task. At test time, their model can infer the best potential sequence of motifs to execute to solve a novel instance of a given task. They apply their model to (abstract) rule learning and motor learning.

**Strengths:**

1. The paper is clearly structured and well written.

2. The authors tackle a timely issue for intelligent systems. Proposing novel methods that can improve the ability of AI models to adapt quickly and learn from fewer examples by re-using previous motifs is an interesting, alternative direction to brute force scaling, which has been the leitmotif of bigtechs. I believe more research should go towards this direction.

**Weaknesses:**

Although I am extremely sympathetic with the type of work proposed by the authors, I have three main concerns with the current state of the paper. I will delineate these concerns below not any order of importance:

1. The authors do not properly position their paper, and a comprehensive comparison with previous methods is lacking. In particular, I would greatly encourage the authors to have a look at the following papers: Zhen et al. (2024 - https://www.nature.com/articles/s41467-024-52289-3), Hurtado et al. (2021 - https://proceedings.neurips.cc/paper/2021/file/761e6675f9e54673cc778e7fdb2823d2-Paper.pdf), Hummos (2023 - https://arxiv.org/abs/2205.11713) , and perhaps more crucially Hummos et al. (2024 - https://arxiv.org/pdf/2407.17356). Such papers are extremely closely related to the proposal of the authors, especially the last one, which also proposes to cast meta-learning as a probabilistic inference problem. Moreover, Hummos et al. (2024) provides a much more detailed description of the link between their model and probabilistic inference. In sum, I believe that the current version of the proposed manuscript should substantially improve positioning the paper with respect to previous and related work.

2. The authors assume a certain temporal structure in the tasks, and propose adaptive behavior as recombining “closed” learned motifs in a specific sequential order. Whereas this might make sense in the tasks they propose, the usefulness of compositional learning far extends this setting, and is particularly powerful when you can select which “parts” of these motifs need to be recombined to solve a novel task. For instance, if I learn to play football, I learn to infer that the goal of this sport is to put the ball in the goals. If I learn to play basketball, I learn that I have to play with my hands and put the ball in the net. Now, if I see people playing handball, I can directly infer that the rule is to put the ball in the goals. Here, inferring the goal of handball is not tied to sequentially structuring the motifs of basketball and football, but rather to re-combing parts of these motifs. Such recombinations are tackled in Hummos et al. (2024). In contrast, the authors propose a recombination that has been tackled in previous work (see Logiaco et al., 2021, Cell reports). I do however acknowledge that in this work, the authors do not require task IDs for flexible recombination, similar to Hummos (2023, ICLR).

3. The authors assess their model in simplified “toy” tasks that can hardly convince on the usefulness of this approach at the practical scale. Whereas I typically do not “condemn” papers for using toy tasks, similar approaches have shown value in vision (CIFAR-10) and language (BabyLM challenge) (see Hummos et al., 2024), which sets an expectation over what tasks would (at the very least) be evaluated by such proposals. Indeed, whereas gating mechanisms are useful in these toy settings, they are notoriously hard to scale when backbone models increase in parameter size of architecture complexity.

**Questions:**

1. Can the authors explain how their work differs from that of Hummos et al. (2024)? In particular, highlighting why their proposed method would lead to potentially more robust results in specific tasks? It would be interesting to compare both models, and understand if and in which settings one model would outperform the other.

2. Have the authors tried to apply their model to tasks that could show promise at the practical scale (e.g., language modeling?).

3. The graphs in figure 2 and figure 4, are particularly cryptic. It would be good if some effort is put into unfolding them with more clarity. Could the authors describe the figure with “pointers”  and add the x and y dimension labels when there are none?

---

> ### Author Response · Authors · 2025-11-20
> **Response (1/5)**
>
> We thank the reviewer for their careful reading and extensive feedback. We appreciate all the pointers to relevant literature and apologise for missing those previously. Embedding the current work into existing literature is important to us, as we consider this paper somewhat of an interdisciplinary contribution: a conceptually new approach to metalearning in machine learning, but also a model for composition and reuse in cognitive science, and a neural model of modularity and sequence inference in neuroscience. We’ve incorporated the reviewer’s remarks and believe the positioning of the paper is now much improved.
>
> The reviewer’s comments also made us realise we haven’t drawn the contrast to previous work sharply enough. While the references are all closely related, we don’t think any other work solves the meta-learning problem like we do: 1) simultaneously learning modules and their sequencing statistics across training tasks, 2) as a probabilistic generative model of tasks without task identity input, 3) to infer a test task’s solution through probabilistic inference without parameter updates. We think our approach has conceptual (e.g. the explicit definition of learning timescales: slow synaptic learning of general training task properties, fast inferential learning of the correct module sequence to perform a test task) and functional (e.g. once the rules for sequencing modules are learned, they automatically extend to longer sequences) advantages. We have now further clarified these differences with other approaches, with additional simulations in support, and thank the reviewer for prompting us to do so. We have included the changes and responses in more detail below.
>
> >The paper is clearly structured and well written. The authors tackle a timely issue for intelligent systems. Proposing novel methods that can improve the ability of AI models to adapt quickly and learn from fewer examples by re-using previous motifs is an interesting, alternative direction to brute force scaling, which has been the leitmotif of bigtechs. I believe more research should go towards this direction.
>
> We’re glad the reviewer finds the paper clearly structured and well written. We’d also like to echo the reviewer’s sentiment here that rapid adaptation and few-shot learning could greatly benefit from structured reuse. We believe that such compositional approaches are an exciting direction to explore - both to improve AI models, and to understand how the brain can rapidly solve problems it hasn’t encountered before.
>
> > Although I am extremely sympathetic with the type of work proposed by the authors, I have three main concerns with the current state of the paper. I will delineate these concerns below not any order of importance:
>
> We thank the reviewer for their comments. They have been extremely valuable both for positioning the paper in a broader literature and emphasising where the novelty of our contribution lies. We have addressed the comments in two ways: 1) by revising the discussion to include the relevant references, and 2) by adding simulations and text edits that demonstrate the advantages of our approach. We hope the reviewer agrees that these majorly strengthen the paper.
>
> > The authors do not properly position their paper, and a comprehensive comparison with previous methods is lacking. In particular, I would greatly encourage the authors to have a look at the following papers: Zhen et al. (2024 - https://www.nature.com/articles/s41467-024-52289-3), Hurtado et al. (2021 - https://proceedings.neurips.cc/paper/2021/file/761e6675f9e54673cc778e7fdb2823d2-Paper.pdf), Hummos (2023 - https://arxiv.org/abs/2205.11713) , and perhaps more crucially Hummos et al. (2024 - https://arxiv.org/pdf/2407.17356). Such papers are extremely closely related to the proposal of the authors, especially the last one, which also proposes to cast meta-learning as a probabilistic inference problem. Moreover, Hummos et al. (2024) provides a much more detailed description of the link between their model and probabilistic inference. In sum, I believe that the current version of the proposed manuscript should substantially improve positioning the paper with respect to previous and related work.
>
> Thanks very much for pointing out these references. We have revised the discussion to emphasise this body of work, and how it relates to our method. It is particularly encouraging that many of these papers are strongly grounded in (biological) neural substrates. This is something we’re keen to explore in relation to our architecture, and we think the thalamic gating of prefrontal activity is a promising candidate mechanism for implementing gated modular computation in the brain.
>
> [CONTINUED BELOW]

---

> > ### Author Response · Authors · 2025-11-20
> > **Response (2/5)**
> >
> > We also want to use this opportunity to draw a sharper contrast between these methods and our approach. There are two important differences that we’d like to highlight. First, inferring a test task after having learned the generative model across training tasks requires a single trial/episode and no weight updates at all in our framework. Second, by learning sequencing statistics across training tasks, rather than a static/task-id-dependent gating, we can generalise beyond training task length. We’ll further expand on these two claims with additional simulations below. The referenced other approaches do not share both these properties.
> >
> > Hurtado et al., 2021 learn a set of reusable modules that are gated by input-dependent masks to solve new tasks. This improves learning new tasks because it reuses knowledge from previous tasks, but doesn’t consider the problem of sequencing modules for timeseries data, and requires task identity during training and weight updates on new tasks. Zhen et al., 2024 propose a model where mediodorsal (MD) thalamus infers context from prefrontal cortex activations (PFC) and gates prefrontal clusters. This model exhibits modular computation and continual learning in a brain-inspired architecture, but also requires weight updates to learn new tasks, and doesn’t learn how modules are sequenced as MD is not recurrent. Hummos 2023 does learn new tasks without weight updates. The paper elegantly separates learning regimes, analogous to synaptic versus inferential learning in our framework: gradient updates on network parameters during training, and gradient updates on a latent embedding that multiplicatively gates network units on test tasks. This is different from our probabilistic inference of test task solutions in two ways. First, it can take many latent updates before the test task is solved, because the gradient steps on latent embedding vectors don’t implement optimal probabilistic inference. Second, the latent is static within a trial, and can therefore only mix together previous tasks, but not go beyond them. This will not generalise to test tasks that are longer than training tasks, as we’ll show below. We agree with the reviewer that Hummos et al., 2024 is the most related to our implementation. They introduce a single-step update to the latent embedding vector which circumvents the first issue mentioned above for Hummos 2023. But the second issue still stands: the model rapidly infers embedding vectors but can’t sequence them. This can provide ‘static’ compositionality - the simultaneous recombination of task elements, like the reviewer’s “football + basketball = handball” example or “MemoryPro + AntiGo = MemoryAnti” in Driscoll et al., 2024, which is an extremely useful capacity - but will not learn the general rules for sequencing modules, like the ‘task grammar’ learned by our gating network. We are excited to see these developments in parallel to our efforts, which each will have their own strengths, and apologise for missing them originally.
> >
> > In the new simulations below we’ll first demonstrate that our model infers test task solutions without weight updates, whereas this is not true for standard ML gradient-based meta-learning. Here the control model is an RNN that receives task identity input on the training and test tasks. The green line, for which the recurrent weights are frozen and input weights retrained like in Driscoll et al., 2024, is of particular interest because it achieves effectively the same as Hummos 2023/Hummos et al., 2024, except by optimising weights rather than the latent embedding vector directly.
> >
> >  [FIGURE 3E]
> >
> > But when we retrain on test tasks that are twice as long as the train tasks, learning new input weights is no longer sufficient, as the temporal composition is beyond those seen in the train tasks. This is not just about learning speed - it’s about asymptotic performance.
> >
> > [FIGURE 3F]
> >
> > We hope that these additional comparisons and results clarify how the different models fit together, but would be very happy to discuss follow-up questions here.
> >
> > [CONTINUED BELOW]

---

> > > ### Author Response · Authors · 2025-11-20
> > > **Response (3/5)**
> > >
> > > We have added the panels above to Fig 3 with the following text in the manuscript:
> > >
> > > _“To demonstrate the effectiveness of single-episode test task inference, we compare it to the common alternative: learning to perform the test task by updating model parameters through gradient descent (Figure 3e). We use a RNN with task identity input as in Figure 3b that takes a gradient descent step after each test task episode. The learning curves show clear signs of reuse: the model that learns the test task from scratch (blue) learns slower than any of the other models that have been pre-trained to perform the training tasks first. However, whether we pre-trained with standard gradient descent (orange) or model-agnostic meta-learning (red) or meta-learning for domain generalisation (purple) doesn't make a difference. All learn on a timescale of hundreds of episodes which is qualitatively different from the single-episode inference (grey). The fact that only relearning the input weights of the pre-trained model (green) performs equally well indicates that the recurrent dynamics from pre-training are sufficient to solve the test task, and the network only needs to learn to map the test tasks onto these dynamics. This is no longer the case if the test task is longer than the training tasks (Figure 3f). Freezing the recurrent weights and retraining input weights (green) then doesn't reach the same asymptotic performance as (re)learning all weights (blue, orange). Because our model learns the general rules for sequencing modules, it automatically generalises to longer tasks without retraining (grey).”_
> > >
> > > And we have added the following discussion paragraph to the manuscript (we cite Hurtado et al., 2021 in the preceding discussion paragraph):
> > >
> > > _“Solving new tasks without parameter updates is also related to in-activity (Hochreiter et al., 2001), in-memory (Santoro et al., 2016), and in-context (Brown et al., 2020; Oswald et al., 2023)learning, but those lack modularity and probabilistic reasoning - both key for compositionality in our framework. Indeed, the approach in Hummos et al. 2024 relies on compositional inference to achieve meta-learning without parameter updates, making it particularly closely related to ours. It proposes an efficient gradient update on a latent task embedding to rapidly discover new embeddings for test tasks. This allows for combining learned computations into new solutions by optimising activity (instead of weights as in Driscoll et al., 2024 or the green line in Figure 3e), but not for sequencing modules along learned transition statistics. Importantly, Hummos 2022 connects these latent embedding vectors to neural processing in the brain, by interpreting them as mediodorsal thalamic neurons that multiplicatively gate prefrontal cortex. This thalamic gating mechanism also offers a potential (biological) neural substrate for our sequential activation of modules, as it could gate different clusters of recurrent neurons in different temporal contexts (Zheng et al., 2024). Indeed, thalamic gating can be used to flexibly sequence cortical motifs to generate motor outputs (Logiaco et al., 2021). Together with the finding that motor learning is well-explained by probabilistic contextual inference (Heald et al., 2021), these studies draw intriguing parallels between our model and brain computation.”_
> > >
> > > > The authors assume a certain temporal structure in the tasks, and propose adaptive behavior as recombining “closed” learned motifs in a specific sequential order. Whereas this might make sense in the tasks they propose, the usefulness of compositional learning far extends this setting, and is particularly powerful when you can select which “parts” of these motifs need to be recombined to solve a novel task. For instance, if I learn to play football, I learn to infer that the goal of this sport is to put the ball in the goals. If I learn to play basketball, I learn that I have to play with my hands and put the ball in the net. Now, if I see people playing handball, I can directly infer that the rule is to put the ball in the goals. Here, inferring the goal of handball is not tied to sequentially structuring the motifs of basketball and football, but rather to re-combing parts of these motifs. Such recombinations are tackled in Hummos et al. (2024). In contrast, the authors propose a recombination that has been tackled in previous work (see Logiaco et al., 2021, Cell reports). I do however acknowledge that in this work, the authors do not require task IDs for flexible recombination, similar to Hummos (2023, ICLR).
> > >
> > > [CONTINUED BELOW]

---

> > > > ### Author Response · Authors · 2025-11-20
> > > > **Response (4/5)**
> > > >
> > > > We thank the reviewer for raising this issue, as it allowed us to better clarify the novelty of our model. We agree wholeheartedly that compositionality is a powerful mechanism that can lead to advantages in many areas of cognition, and that temporal composition is one of these. But it’s an important one, as many real-world tasks unfold through time, and we disagree that the flexible combination of recurrent motifs in a sequence has already been solved. Indeed, the work by Logiaco et al., 2021 is a major contribution to this understanding. That paper explores how motifs can be temporally sequenced together. Crucially, however, in that work the identity of each motif in the sequence is supplied to the network at the appropriate time (in the form of proper preparatory input signal). There are also no regularities in the temporal structure of these motifs. The toy task in Hummos et al., 2024 is set up similarly. The network is explicitly provided with the task abstraction vector during training, and doesn’t learn about its transition statistics. By contrast, our model simultaneously learns the motifs and their sequencing statistics without any explicit instruction. This allows it to infer both the identity and timing of module switches in test tasks even when these switches are uncued. In our view this is a natural and unresolved question in the compositionality literature, and our model offers a novel solution.
> > > >
> > > > To be explicit in this response, the novelty of our solution has two sources. First, we separately learn dynamics within modules from the dynamics between modules across training tasks without task identity inputs. Second, by relying on the learned between-module dynamics, we can infer module sequences without parameter updates even in the absence of feedback and beyond training task length. This is a problem definition that classical gradient-based meta-learning methods struggle with (see the first figure above), but isn’t easily solved by compositional methods either (see the second figure above). These new results show that the proposal to pre-train RNN dynamics across training tasks and subsequently relearn input weights on test tasks (Driscoll et al., 2024) can’t capture sequences longer than those in the training tasks. We expect this will be identical for methods that infer latent task abstractions (Hummos 2023, Hummos et al., 2024), as they achieve effectively the same as relearning input weights - although with activation instead of weight changes, and in fewer updates. That is because all these methods rely on the type of compositionality mentioned by the reviewer (“football + basketball = handball” or “MemoryPro + AntiGo = MemoryAnti”), while our “grammar of tasks” is more appropriate for sequential task building.
> > > >
> > > > > The authors assess their model in simplified “toy” tasks that can hardly convince on the usefulness of this approach at the practical scale. Whereas I typically do not “condemn” papers for using toy tasks, similar approaches have shown value in vision (CIFAR-10) and language (BabyLM challenge) (see Hummos et al., 2024), which sets an expectation over what tasks would (at the very least) be evaluated by such proposals. Indeed, whereas gating mechanisms are useful in these toy settings, they are notoriously hard to scale when backbone models increase in parameter size of architecture complexity.
> > > >
> > > > We thank the reviewer for this remark. We agree that we apply our model in “toy” tasks, but believe that for the current proof-of-principle result, these are extremely valuable. Even though these tasks are synthetic and low-dimensional, they have three attractive properties: 1) they are hard: they require simultaneously learning arbitrary data transformations and how these are sequenced from uninformative input, 2) they are controlled: we know the true modules and sequences so can verify what the model actually learns, 3) they demonstrate application in different domains: rule learning and motor learning. We think this is a good balance for the current paper, and that in its current scale it could already be applied to e.g. cognitive modelling of reusable computation. We have added this as a discussion point:
> > > >
> > > > _“The tasks that we designed to test our model are similarly proof-of-principle. Despite being synthetic and low-dimensional, these tasks are valuable because 1) they are hard: they require simultaneously learning arbitrary transformations and their sequencing from uninformative input, 2) they are controlled: we know the true modules and sequences so can verify what the model actually learns, 3) they demonstrate application in different domains: rule learning and motor learning. Moreover, the model's core ideas, namely slow synaptic learning of modules and their transitions across training tasks and fast inferential learning of new module combinations on test tasks, will apply to any problem with sequential modular structure.”_
> > > >
> > > > [CONTINUED BELOW]

---

> > > > > ### Author Response · Authors · 2025-11-20
> > > > > **Response (5/5)**
> > > > >
> > > > > > Can the authors explain how their work differs from that of Hummos et al. (2024)? In particular, highlighting why their proposed method would lead to potentially more robust results in specific tasks? It would be interesting to compare both models, and understand if and in which settings one model would outperform the other.
> > > > >
> > > > > We hope that we have clarified this more in the answers above. One way that we find useful to think about this, is that for Hummos et al., 2024 a task is a single point in a continuous latent space. The latent space can be learned across training tasks, so that inferring the latent embedding vector for a test task recombines elements from the training tasks in new configurations. Our tasks on the other hand are trajectories through discrete states (i.e. module activations). The states (modules) and their transitions (gating) are learned from training tasks, so that inferring the trajectory for a test task recombines elements in new sequences. We therefore believe that the outcomes of model comparisons will depend on the nature of composition in the task. When there’s a lot to gain by static composition, Hummos et al., 2024 and related methods will likely perform well. When the task is mainly characterised by dynamic composition, through structured sequencing of a set of reusable modules, we expect our model to be at its strongest.
> > > > >
> > > > > > Have the authors tried to apply their model to tasks that could show promise at the practical scale (e.g., language modeling?).
> > > > >
> > > > > As explained above, we think that for a proof-of-principle it’s most important that we can mechanistically understand how the model is solving the task. That’s most convincingly demonstrated when we know the ground truth modules and sequences. We see potential for applications in domains with modular and regular sequential structure, and can imagine language modelling being one of those, but have not tried that in the current paper.
> > > > >
> > > > > > The graphs in figure 2 and figure 4, are particularly cryptic. It would be good if some effort is put into unfolding them with more clarity. Could the authors describe the figure with “pointers” and add the x and y dimension labels when there are none?
> > > > >
> > > > > We completely agree. We have updated the figure panels with more pointers and labels to improve their clarity:
> > > > >
> > > > > [FIGURE 2, 4]

---

> > > > > > ### Comment · Reviewer_Gbhf · 2025-11-25
> > > > > >
> > > > > > I thank the authors for their thourough review and clear responses. Some of my concerns were addressed. In particular, the paper now is better positioned. However, the novelty over Hummos et al. (2024) does not seems stark enough to warrent an accept in the current form. It would have been nice to observe empirical results in "toy tasks" that are more related to real-world tasks. In light of the replies I will raise my score to 4, and thank again the authors for their thorough responses.

---

> > > > > > > ### Author Response · Authors · 2025-12-02
> > > > > > > **Response (1/1)**
> > > > > > >
> > > > > > > We thank the reviewer for their response. We’re glad they appreciate our revisions and are grateful for the updated score. We respectfully disagree with the lack of novelty over Hummos et al., 2024. To compactly summarise the arguments laid out above, our approach tackles (1) a new problem in (2) a new way to (3) achieve new results:
> > > > > > >
> > > > > > > 1. We consider tasks that are new temporal compositions of a discrete set of reusable modules along structured transitions, whereas Hummos et al., 2024 finds static compositions of continuous task abstractions.
> > > > > > > 2. We solve test tasks by probabilistic inference on a learned generative model, trained by backpropagation through a particle filter, whereas Hummos et al., 2024 calculates gradients on task abstraction vectors.
> > > > > > > 3. We infer solutions from sparse feedback and beyond training task length, which is not feasible for Hummos et al., 2024. This is a key feature in terms of generalisation. To use an analogy in natural languages, this is the difference between a language model that can deal with sentences of say 10 words only, and one that can deal with sentences of arbitrary length.
> > > > > > >
> > > > > > > Thanks again for the extensive engagement - we believe the reviewer’s feedback has majorly improved the manuscript.

---

### Official Review · Reviewer_hQoD · 2025-10-31

**Soundness:** 2
**Presentation:** 2
**Contribution:** 2
**Rating:** 2
**Confidence:** 2

**Summary:**

The paper presents a new proof of principle for compositional meta-learning through probabilitistic task inference. The underlying design choices are studied and compared to a base architecture.

**Strengths:**

The paper actually cites references from before the deep learning revolution which is nice to read for once.

The paper is well written and easy to read. The proposal seems technically sound.

In its style it is for sure an unusual paper (which could be something positive or negative).

**Weaknesses:**

The paper seems to be a very limited proof of concept. While there are claims that other approaches can not solve these tasks e.g. "This is in stark contrast to traditional meta-learning approaches  ..." i.e there i no baseline comparison to any other approach.

It would indeed be nice to see that no other approach is able to solve the tasks beyond just stating it. At least to me that is not obvious.

Or if it is indeed a proof of principle for one model class then at least it should be scaled up. As it is it sounds very narrow in its placement for readership.

**Questions:**

What are the precise claims that other meta learning approaches can not do? Is it accurate posteriors? Is it the tasks? What precisely is impossible to solve with SOTA meta learning approaches? The studied dimension of the examples are not huge, is it a data problem, a zero-shot problem or what precisley is the bottleneck for existing approaches.

---

> ### Author Response · Authors · 2025-11-20
> **Response (1/3)**
>
> We thank the reviewer for their assessment. We’re glad they liked our approach and presentation. We do feel we failed to convey why our method is different from existing approaches, and how and when it outperforms those. We apologise for not being explicit enough about this in our original manuscript. We have now updated the text and we’ve run additional simulations to address the reviewer’s comments, which we think majorly improved the paper. These are described in more detail below. If anything remains unclear, we’d be very happy to provide further explanation/edits here!
>
> > The paper actually cites references from before the deep learning revolution which is nice to read for once. The paper is well written and easy to read. The proposal seems technically sound. In its style it is for sure an unusual paper (which could be something positive or negative).
>
> We are happy that the reviewer found the paper well-written and easy to read. We can understand why the reviewer notices the “unusual style” of the paper, as it’s positioned somewhat between research fields. As we see it, it’s an important conceptual development with several proofs-of-principle for metalearning in ML, but we also believe it addresses questions about composition and reuse from cognitive science, and provides a neural model of modularity and inference that is relevant for neuroscience. Because of this interdisciplinary motivation we draw inspiration from multiple bodies of literature, and the paper may not be structured like a typical ML paper. But we believe this interface of machine learning, cognitive science, and neuroscience is an exciting space to explore; we hope that the reviewer agrees.
>
> > The paper seems to be a very limited proof of concept. While there are claims that other approaches can not solve these tasks e.g. "This is in stark contrast to traditional meta-learning approaches ..." i.e there is no baseline comparison to any other approach.
> It would indeed be nice to see that no other approach is able to solve the tasks beyond just stating it. At least to me that is not obvious. Or if it is indeed a proof of principle for one model class then at least it should be scaled up. As it is it sounds very narrow in its placement for readership.
>
> Thanks for these comments. We now realise we have not been explicit enough in contrasting our work with previous meta-learning approaches. There is one very important conceptual difference, which really changes the problem definition: we infer new tasks without parameter updates. This very explicitly separates two different types of learning. There is slow, synaptic learning through gradient descent to learn the parameters of module and gating networks across many training tasks. Then there is fast, inferential learning to find the best sequence of modules given a test task - without changing any model parameters. Traditional meta-learning approaches (like MAML, Reptile, MLDG) don’t have this inferential learning, but instead also learn new tasks through weight updates with gradient descent. Their goal is to find parameters so that there are only small weight updates required on new tasks. Ours is to not have any weight updates at all.
>
> We don’t think we made this point clearly enough in the original manuscript, so we thank the reviewer for making us aware of this. Previously, we only showed comparisons on performance after a single episode of a new task (Fig 3). We now added a simulation of learning through gradient descent over many episodes of the new task (Fig 3e). This shows that having learned previous tasks speeds up new task acquisition, but our architecture is the only one that solves new tasks immediately. The fact that learning the training task through meta-learning methods (MAML, MLDG) doesn't outperform learning those by standard gradient descent shows the difficulty of our setting for current methods. Intuitively, there is no universal set of parameters which reside close to a solution across tasks. Instead to learn quickly, you need a different kind of learning altogether - one which knows about task structure and can make inferences on this structure in a probabilistic way. But the main takeaway from this new panel is the qualitatively different time scales of test task acquisition: a few hundred episodes versus one.
>
> [FIGURE 3E]
>
> [CONTINUED BELOW]

---

> > ### Author Response · Authors · 2025-11-20
> > **Response (2/3)**
> >
> > This is only possible because we propose a compositional architecture, with a discrete set of reusable modules, so that we can learn a probabilistic generative model across training tasks. Then during inference, we compose new task computations from the learned modules (as ‘task syllables’) in accordance with their learned statistics (as ‘task grammar’). Because the model extracts the rules by which tasks are constructed, it can even meta-learn test tasks that are far beyond the training set, e.g. twice as long as any previously observed task in Fig 3f. This is not the case for other methods that freeze the recurrent weight but retrain input weights on new tasks (green), which is similar to the approach in Driscoll et al., 2024 or Hummos 2023, Hummos et al., 2024.
> >
> > [FIGURE 3F]
> >
> > We have included these new panels in Figure 3, and added the following text to the manuscript:
> >
> > _“To demonstrate the effectiveness of single-episode test task inference, we compare it to the common alternative: learning to perform the test task by updating model parameters through gradient descent (Figure 3e). We use a RNN with task identity input as in Figure 3b that takes a gradient descent step after each test task episode. The learning curves show clear signs of reuse: the model that learns the test task from scratch (blue) learns slower than any of the other models that have been pre-trained to perform the training tasks first. However, whether we pre-trained with standard gradient descent (orange) or model-agnostic meta-learning (red) or meta-learning for domain generalisation (purple) doesn't make a difference. All learn on a timescale of hundreds of episodes which is qualitatively different from the single-episode inference (grey). The fact that only relearning the input weights of the pre-trained model (green) performs equally well indicates that the recurrent dynamics from pre-training are sufficient to solve the test task, and the network only needs to learn to map the test tasks onto these dynamics. This is no longer the case if the test task is longer than the training tasks (Figure 3f). Freezing the recurrent weights and retraining input weights (green) then doesn't reach the same asymptotic performance as (re)learning all weights (blue, orange). Because our model learns the general rules for sequencing modules, it automatically generalises to longer tasks without retraining (grey).”_
> >
> > So while we agree this is a proof-of-principle, we think our architecture solves new tasks in a qualitatively different way, instead of outperforming SOTA by some particular margin. The tasks that we use to evaluate the model are synthetic and low-dimensional, but we think they are useful for three reasons: 1) they are hard: they require simultaneously learning arbitrary data transformations and how these are sequenced from uninformative input, 2) they are controlled: we know the true modules and sequences so can verify what the model actually learns, 3) they demonstrate application in different domains: rule learning and motor learning. It’s not directly obvious to us what scaling up would currently add. We therefore hope that the current set of tasks, with the addition of the explicit comparisons to alternative approaches above, addresses the reviewers concern about the paper’s scope. But please do let us know if anything remains unclear!
> >
> > > What are the precise claims that other meta learning approaches can not do? Is it accurate posteriors? Is it the tasks? What precisely is impossible to solve with SOTA meta learning approaches? The studied dimension of the examples are not huge, is it a data problem, a zero-shot problem or what precisley is the bottleneck for existing approaches.
> >
> > We hope that the above clarifies these questions. Thanks again for asking them, because they very much helped us improve the clarity of our claims. Other meta-learning approaches cannot learn new tasks without weight updates, because parameter updates are the only way in which they can learn. We can, because we learn a probabilistic generative model, which supports a different way of learning from data: parameter-update-free inference. This inference process also naturally provides interpretable posteriors and likelihoods, which other methods won’t do unless specifically trained to. The reason why is architectural. By defining our architecture as a probabilistic generative model, trained by backpropagation through a particle filter, we can benefit from powerful and data-efficient probabilistic methods that other methods don’t support.
> >
> > [CONTINUED BELOW]

---

> > > ### Author Response · Authors · 2025-11-20
> > > **Response (3/3)**
> > >
> > > We have added this clarification to the discussion:
> > >
> > > _“This is in stark contrast to traditional meta-learning approaches that explicitly focus on weight up dates. Finn et al. (2017); Li et al. (2018); Nichol et al. (2018) adjust the training loop to learn parameters that require minimal gradient-based adaptation on new tasks. But exactly because they rely on parameter updates, they meta-learn qualitatively slower than our single-episode task inference. This inference is afforded by a modular architecture (Pfeiffer et al., 2024), which has been linked to compositional generalisation (Chang et al., 2018) as well as resource efficiency (Shazeer et al., 2017; Fedus et al., 2022).”_
> > >
> > > And explained our rationale for the tasks and core model features:
> > >
> > > _“The tasks that we designed to test our model are similarly proof-of-principle. Despite being synthetic and low-dimensional, these tasks are valuable because 1) they are hard: they require simultaneously learning arbitrary transformations and their sequencing from uninformative input, 2) they are controlled: we know the true modules and sequences so can verify what the model actually learns, 3) they demonstrate application in different domains: rule learning and motor learning. Moreover, the model's core ideas, namely slow synaptic learning of modules and their transitions across training tasks and fast inferential learning of new module combinations on test tasks, will apply to any problem with sequential modular structure.”_

---

### Official Review · Reviewer_6GAZ · 2025-10-31

**Soundness:** 4
**Presentation:** 3
**Contribution:** 3
**Rating:** 8
**Confidence:** 3

**Summary:**

The paper proposes a proof of principle for a novel compositional meta-learning approach that uses a gating mechanism to train, sequentially select and combine different experts. The paper demonstrates its effectiveness in two cases: (1) rule learning and (2) motor learning tasks. It performs well, especially in scenarios with sparse feedback signals, and generalizes effectively to unseen test data with longer time horizons compared to the training data.

**Strengths:**

- The proposed method is novel, and the experiments are well done and show excellent results. It shows that the proposed method can perfectly recover individual tasks, and their composition achieves strong performance on the overall tasks.
- The paper is very well written. The supplementary code is also well prepared, demonstrating the paper’s reproducibility.

**Weaknesses:**

- The main weakness of the paper, is that as a proof-of-principle paper, the bigger picture is not emphasized enough. Specifically, what main principles it proposes when considering architectures other than LSTMs. For example, which parts of the proposed method are transferable to other architectures? say a Transformers for example. However, I think this is more a matter of clarity rather than a fundamental flaw.
- The number of experiments is somewhat limited, but for a proof-of-principle paper, the experiments presented are well executed and thoroughly analyzed. So, while this could be seen as a weakness, I don’t think it diminishes the overall value of the paper.

**Questions:**

__Q1.__ Could you comment on the scenario where the number of expert modules or the operation chunk is misspecified? For instance, when the number of expert modules is substantially smaller than the number of true operations in the dataset? How crucial is this for the method’s effectiveness?

__Q2.__ The main limitation of the amortized method, despite its advantage of not requiring parameter updates when new datasets arrive, is its tendency to perform poorly on out-of-distribution test data. Could you comment on how this issue might be addressed when using the proposed method? For instance, is it possible or straightforward to detect test data that lie completely outside the composition of the training data using this method (or a potential extension of it)?

__Q3.__ In Figure 3b, how is the task input specified? During testing, does the experiment include task inputs that were not seen during training? If that’s the case, what about introducing a “wild card token” task input during training, which is assigned to a random task, and then using it during the testing phase on new unseen test data?

---

> ### Author Response · Authors · 2025-11-20
> **Response (1/2)**
>
> We’d like to thank the reviewer for their careful evaluation of the manuscript, and specifically for the suggestions in the “questions” section - these are all extremely relevant and improve our paper both in its details and in its broader scope. Thanks very much for the constructive reviewing. We will respond to these suggestions, and the other remarks, below.
>
> > The proposed method is novel, and the experiments are well done and show excellent results. It shows that the proposed method can perfectly recover individual tasks, and their composition achieves strong performance on the overall tasks.
> The paper is very well written. The supplementary code is also well prepared, demonstrating the paper’s reproducibility.
>
> We are happy to hear that the reviewer feels the paper was very well written, and appreciate their interpretation of the method and results.
>
> > The main weakness of the paper, is that as a proof-of-principle paper, the bigger picture is not emphasized enough. Specifically, what main principles it proposes when considering architectures other than LSTMs. For example, which parts of the proposed method are transferable to other architectures? say a Transformers for example. However, I think this is more a matter of clarity rather than a fundamental flaw.
>
> We thank the reviewer for their comment. We agree that this is a proof-of-principle paper so it’s important that the bigger picture is clear. The core principles that our model relies on in the most abstract sense is 1) the slow (synaptic) learning of reusable modules and their combination rules, and 2) the fast (inferential) learning of new combinations to solve new tasks. In the bigger picture, these principles can be applied to a wide range of tasks, on a wide range of architectures. In terms of tasks, these principles could support complex motor skill acquisition (learning to serve in tennis combines throw, jump, and hit actions) or planning (cooking a new dish combines operations like chopping, frying, boiling) or reasoning (understanding which sequence of functions causes the behaviour of a particular piece of code). Moreover, we hope to apply these principles to empirical results of modular computation in the brain in future work. In terms of architecture, both the gating RNN and module RNNs can be swapped out for any other function, including parameterised probability distributions or neural network architectures. Specifically, we expect that replacing the gating network by a Transformer would boost the representation of sequence regularities, so that it could model task structure akin to the complicated hierarchical grammars that govern natural language.
>
> We have added the following to the discussion to emphasise the bigger picture:
>
> _“The tasks that we designed to test our model are similarly proof-of-principle. Despite being synthetic and low-dimensional, these tasks are valuable because 1) they are hard: they require simultaneously learning arbitrary transformations and their sequencing from uninformative input, 2) they are controlled: we know the true modules and sequences so can verify what the model actually learns, 3) they demonstrate application in different domains: rule learning and motor learning. Moreover, the model's core ideas, namely slow synaptic learning of modules and their transitions across training tasks and fast inferential learning of new module combinations on test tasks, will apply to any problem with sequential modular structure. In addition to task domains, we note that our architecture also generalises to different gating and module components. Replacing the gating RNN by a modern transformer would allow for learning complex task grammars not unlike the one governing natural languages (Vaswani et al., 2017). Forcing the module RNNs to be low rank would implement a bias towards combinations of simple operations through low-dimensional dynamics (Mastrogiuseppe and Ostojic, 2018). Together, our results and suggestions chart a path towards the rapid composition of new solutions from learned elements across domains.”_
>
> > The number of experiments is somewhat limited, but for a proof-of-principle paper, the experiments presented are well executed and thoroughly analyzed. So, while this could be seen as a weakness, I don’t think it diminishes the overall value of the paper.
>
> We thank the reviewer for this assessment, and agree on the value of carefully executed and analysed experiments. We believe that our approach of having synthetic tasks where we know the underlying regularities is extremely important to validate the model. By having full control over what the true modules are, and how their sequences are structured, we can verify that the model learns those. We show the method’s generality by applying it in the rule learning and motor learning domain.
>
> [CONTINUED BELOW]

---

> > ### Author Response · Authors · 2025-11-20
> > **Response (2/2)**
> >
> > > Q1. Could you comment on the scenario where the number of expert modules or the operation chunk is misspecified? For instance, when the number of expert modules is substantially smaller than the number of true operations in the dataset? How crucial is this for the method’s effectiveness?
> >
> > Thanks - that is a great suggestion. When the number of modules is smaller than the number of true operations, and these true operations are sufficiently orthogonal, then our model will fail (by design) and learn approximations/combinations of true operations. When the number of modules is larger than the number of true operations, the optimal solution would be to only use a subset of modules, because that maximises the likelihood of true sequences (but there’s always a risk of getting stuck local optima). For the method’s effectiveness, one approach could therefore be to start with many potential modules and prune. We are now running simulations for both these scenarios and will add them to the supplementary information.
> >
> > > Q2. The main limitation of the amortized method, despite its advantage of not requiring parameter updates when new datasets arrive, is its tendency to perform poorly on out-of-distribution test data. Could you comment on how this issue might be addressed when using the proposed method? For instance, is it possible or straightforward to detect test data that lie completely outside the composition of the training data using this method (or a potential extension of it)?
> >
> > We thank the reviewer for raising this issue. We actually believe this is a major strength of our probabilistic approach: the likelihood is a clear metric for how compatible a particular data episode is with what the model has learned from all the training tasks. In our framework a new task could be incompatible with learned structure for two reasons: 1) it requires a module that wasn’t present in the training tasks, or 2) it follows module transitions that were not present in the training tasks. Either of these would show a poor likelihood of the task’s episodes under our learned model. This would be a straightforward criterion for detecting out-of-distribution test data, which could trigger learning new modules or updating the gating network in the continual learning setting. We are now running simulations for both types of out-of-distribution tasks and will report the results.
> >
> > > Q3. In Figure 3b, how is the task input specified? During testing, does the experiment include task inputs that were not seen during training? If that’s the case, what about introducing a “wild card token” task input during training, which is assigned to a random task, and then using it during the testing phase on new unseen test data?
> >
> > Great point, thanks very much. Currently, during testing, we indeed provide task inputs not seen during training. We completely agree that the “wild card token” is a better implementation (although we don’t think this will fundamentally change the results: it effectively makes sure that the “initial guess” for the input weights of new tasks is better, but won’t solve new tasks without retraining). We are now rerunning the simulations with this change and will update the figure accordingly.

---

> > > ### Author Response · Authors · 2025-11-26
> > > **Follow-up (1/1)**
> > >
> > > We’re just following up to confirm that we have implemented each of the suggestions in the reviewer’s questions. They’ve improved the paper significantly - thanks very much again. We have added Figure A1 with the following caption, and summarise the outcomes for each question below.
> > >
> > > _”Data-model mismatch. Starting from the default rule learning task and model (Figure 2), we vary the model while keeping the same tasks (a-d) and vary the test tasks while keeping the same model (e). a) Shift operations (top) in the rule learning task with six operations, with learned modules (bottom) for a model with access to eight modules. These matrices contain in each row $i$ the output of an operation (top) or module (bottom) given the six-dimensional unit vector $i$ as input, as in Figure 2b. Six of the model's modules learn one ground truth shift operation each, the remaining two modules remain random. b) Ground truth (top) and learned (bottom) transitions for the eight-module model on the six-operation task. Each matrix plots transition probabilities for data operations (top) or model modules (bottom) given multiple repeats of the operation or module, as in Figure 2c. The transitions between the first six learned modules mirror the data operation transitions, whereas the remaining two modules remain unused. c) Shift operations (top) and learned modules (bottom) for a four-module model on the same six-operation task. The modules now learn (approximations of) a subset of operations, here operation $0, 2, 4, 5$, with an apparent preference for operations that repeat more timesteps ($2, 4, 5$). d) Learned transitions (bottom) approximate the transition structure for these modules (top). e) Episode log likelihoods per timestep for in-distribution (top) and out-of-distribution (OoD; middle, bottom) test tasks. Here we use the default (six-module) model and training tasks as in Figure 2, but change the test tasks. If the test tasks are compiled from the same operations as the training tasks, through the same transition statistics, the likelihood of episodes from these held-out tasks is high. If the test tasks use different operations (here: random permutations) with the same transition statistics (middle), or the same operations but different transition statistics (here: random switches; bottom), the log likelihood of episodes is low.”_
> > >
> > > Q1: We have simulated the model with more modules than operations (Fig A1a,b) and the model with fewer modules than operations (Fig A1c,d). We find something very much like we predicted in the response above: in the former case, redundant modules remain unused, and in the latter, the modules approximate a subset of operations. We report this result in the main text:
> > >
> > > _”We have so far assumed an equal number of task operations and model modules, but we still find correspondence in learned modules and transitions when there's a data-model mismatch (Figure A1). If there are more modules than operations, the redundant modules remain unused (Figure A1a,b); if there are fewer modules than operations, the modules approximate a subset of the operations (Figure A1c,d).”_
> > >
> > > Q2: We have calculated the likelihood of held-out test task episodes for the model trained on the rule learning tasks (Fig A1e, top), and then repeated this for test tasks with different modules (Fig A1e, middle) and different transitions (Fig A1e, bottom). As expected, the out-of-distribution episodes have a completely different range of likelihoods, which can provide a valuable signal that some form of relearning would be necessary. We mention this in the main text and the discussion:
> > >
> > > _”On the other hand, if test tasks are incompatible with the learned modules or transition statistics, the episode likelihood will be a clear indicator of out-of-distribution data (Figure A1e) which could trigger retraining.”_
> > >
> > > _”In such a setting the model could dynamically add new modules if inference using the existing modules fails, signalled by low episode likelihood (Figure A1e).”_
> > >
> > > Q3: We have rerun the simulation in panel 3b with the suggested “wildcard” test task identity signal (Fig 3b). While the RNN still can’t solve test tasks without retraining, this change reduced the test task error to be in the same range as the other panels, causing the inset plot in the previous version to be no longer required. We completely agree that this is the correct control and a fairer comparison so we have replaced Fig 3b for this new version.

---

### Official Review · Reviewer_rx9b · 2025-11-01

**Soundness:** 2
**Presentation:** 1
**Contribution:** 2
**Rating:** 4
**Confidence:** 3

**Summary:**

The paper proposed a generative modeling approach to address the meta-learning problem. Inspired from the mixture of experts, the proposed method consists of a gating model that selects which module (e.g., another model) should handle a training sample. In addition, the whole modeling follows a time series, in which the hidden state calculated in a previous sample is used in the calculation of the model selection for the next training sample. Particle filtering is used to infer the parameter of the proposed generative model.  Because the paper is hard to understand, this is what I have been understood about the paper so far.

**Strengths:**

The paper employs the idea of modularity from mixture of experts to address the learning problem in meta-learning. In addition, the modeling relies on recurrent neural networks, which takes the temporal effect into the modeling.

**Weaknesses:**

The current presentation of the paper is hard to understand, causing the difficulty to interpret the contribution of the paper. I will make further comments after having the authors clarify. Please refer to the questions below for further clarification to improve the paper.

**Minor**
Since the paper proposes to decompose tasks, one related paper should also be discussed is: Nguyen, C.C., Do, T., Carneiro, G.. (2021). Probabilistic task modelling for meta-learning. In Conference on Uncertainty in Artificial Intelligence.

**Questions:**

The paper is unclear on how a task is defined at line 77. According to the definition specified in the paper, a task is a data generation process yielding multiple episodes. Could the authors make it clear that $\\{(x\_{t}, y\_{t})\\}\_{t = 1}^{T}$ represents $T$ episodes, in which each episode is an input-output pair, or the whole set is represented as an episode as what has been defined in existing meta-learning? Given the subscript $t$ in the notations, I believe it is the former one. In that case, the ordering of samples within a training task will have a major influence on the inference of the generative model's parameters because the modeling depends on time series assumption. However, it does not mention or discussed.

It is unclear how to get Eqs. (7) and (8). Could the authors provide more details on how to arrive at these equations? In addition, the objective function in learning is often either maximizing likelihood or minimizing loss. Eq. (8) is a likelihood and according to the paper: "is used as the training loss". Should it be the other way around, in which the negative log-likelihood is the training loss?

Are the notations used in section 2,2 completely different from previous sections? Since $x_{t}$ and $y_{t}$ are used in previous sections, using them again with different meaning causes confusion.

The paragraph after Eq. (9) is confusing and requires further clarification. For example, why is suddenly a process of **6 shift operations**  introduced? Why is it **6**?

---

> ### Author Response · Authors · 2025-11-20
> **Response (1/3)**
>
> We thank the reviewer for their remarks and for the opportunity to provide clarifications. We’re happy to see the reviewer appreciates the combination of modularity and recurrence, which we agree are two central components of our meta-learning framework. Briefly, our architecture models timeseries through a sequence of recurrent modules, selected by the recurrent gating network, which together allows for rapidly inferring new module sequences. We apologise for the confusion about the model and task setup caused by the rather condensed manuscript. We will answer the reviewer’s questions below, and have expanded the explanation in the manuscript accordingly, to make it easier to understand for future readers. If anything remains unclear, we’d be very happy to provide further clarifications here!
>
> > Since the paper proposes to decompose tasks, one related paper should also be discussed is: Nguyen, C.C., Do, T., Carneiro, G.. (2021). Probabilistic task modelling for meta-learning. In Conference on Uncertainty in Artificial Intelligence.
>
> Thanks for pointing this out. We agree that this paper is very relevant to our approach, as it learns a probabilistic generative model of tasks and then infers the parameters for new tasks. We now cite it in line 60 when we introduce our probabilistic modelling approach in the introduction, because we believe it follows the same principles.
>
> _"Together, the gating network and the modules thus learn a generative model of the “train” tasks. By casting our framework explicitly as a probabilistic generative model, we can solve new “test” tasks through probabilistic inference (Nguyen et al., 2021)"_
>
> > The paper is unclear on how a task is defined at line 77. According to the definition specified in the paper, a task is a data generation process yielding multiple episodes. Could the authors make it clear that $\lbrace (x\_t,y\_t)\rbrace\_{t=1...T}$ represents $T$ episodes, in which each episode is an input-output pair, or the whole set is represented as an episode as what has been defined in existing meta-learning?
>
> We believe this question reflects the reviewer’s main misunderstanding. We now realise this should have been made much more explicit in the paper, because it’s essential for all the following. $\lbrace (x\_t,y\_t)\rbrace\_{t=1...T}$ does not represent $T$ episodes. It represents a single episode, which is a timeseries of T timesteps, with an input ($x\_t$) - output ($y\_t$) pair at each timestep. We define a task as a process that turns a timeseries of inputs $x\_t$ into a timeseries of outputs $y\_t$, for $t=1…T$ timesteps. By episode we mean one particular instance of such a timeseries for a particular task. Our metalearning objective is therefore to figure out the mapping of sequential inputs to sequential outputs from one episode of a test task, after having been trained on a large set of episodes generated from the training tasks.
>
> Intuitively, this may be easiest to understand in the context of rule learning, where the agent has to learn to provide the right outputs through appropriate operations on the inputs (but we believe it applies in other domains as well). Formally, we define a set of tasks $\lbrace q^{(i)}\rbrace\_{i=1...Q}$ where a task $q^{(i)} = \lbrace f^{(i)}\_t(x\_{1:t})\rbrace\_{t=1...T}$ is a timeseries of output functions $f^{(i)}\_t$ that each depend on the history of the inputs. Task  $q^{(i)}$ generates episodes of timeseries $\lbrace x\_t,y\_t\rbrace\_{t=1...T}$. After learning to perform (i.e. produce correct output sequence given the input sequence) each task in $\lbrace q^{(i)}\rbrace\_{i=1...Q}$, the metalearning goal is to solve task $q^{(Q+1)}$ from as few test task episodes as possible.
>
> We approach this problem by assuming that many real-world tasks are modular, so that there is only a limited number of output functions $f^{(i)}\_t$ which are reused in different orders across tasks. In our architecture, the module RNNs must learn to implement each function $f^{(i)}\_t$ and the gating RNN must learn the rules for sequencing them across training tasks. We then use probabilistic inference to infer the mostly likely sequence of modules for a given episode of a held-out test task.
>
> We hope this clarifies the task definition and modelling approach. We have updated the text accordingly. The manuscript, line 76, now reads:
>
> [CONTINUED BELOW]

---

> > ### Author Response · Authors · 2025-11-20
> > **Response (2/3)**
> >
> > _“Our goal is to rapidly find a solution to a new (or test) task $q^{(Q+1)}$, after having encountered a set of previous (or training) tasks $\lbrace q^{(i)}\rbrace\_{i=1...Q}$. We define a task $q^{(i)}$ as a data-generating process $q^{(i)} = \lbrace f^{(i)}\_t\rbrace\_{t=1...T}$ of $T$ timesteps, where $f^{(i)}\_t$ are arbitrary functions that we need to learn. Given a timeseries of inputs $\lbrace x\_t\rbrace\_{t=1...T}$ (for example, i.i.d. sampled from a Gaussian), the task thus generates output timeseries $\lbrace y\_t=f^{(i)}\_t(x\_{1:t})\rbrace\_{t=1...T}$. We refer to a timeseries of $\lbrace (x\_t,y\_t)\rbrace\_{t=1...T}$ input-output pairs for a particular task as an episode. Our model needs to discover the data-generating process $\lbrace f^{(Q+1)}\_t\rbrace\_{t=1...T}$ to produce the correct output $\lbrace y\_t\rbrace\_{t=1...T}$ given input $\lbrace x\_t\rbrace\_{t=1...T}$ for the test task $q^{(Q+1)}$, from a minimal number of episodes. To achieve that, it needs to learn the commonalities across the training tasks $\lbrace q^{(i)}\rbrace\_{i=1...Q}$, and figure out how to apply those in the test task. Importantly, we assume that many real-world tasks are modular: they generate data through varying combinations of sub-processes. In other words, there is a limited number of modules $f^{(i)}\_t$ that get reused in different orders across tasks. Our model therefore needs to learn two key characteristics of the training tasks. First, it must isolate the modules to learn within-module dynamics; and second, it must extract how they are combined to learn between-module dynamics.”_
> >
> > > Given the subscript t in the notations, I believe it is the former one. In that case, the ordering of samples within a training task will have a major influence on the inference of the generative model's parameters because the modeling depends on time series assumption
> >
> > As explained above, indeed one episode is a timeseries of input-output pairs. This time series assumption is part of the problem definition - which is presumably clearer now in the revised text.
> >
> > > It is unclear how to get Eqs. (7) and (8). Could the authors provide more details on how to arrive at these equations?
> >
> > Thanks for the question. As this is a result from particle filtering, we have added the extended explanation below to Appendix A.2:
> >
> > _“With this particle system in place, Equations 7-8 then specify the marginal likelihood that we use for training the model's parameters. Below, we provide additional details of how to arrive at these equations. They are the particle filter Monte Carlo estimate of the marginal likelihood, obtained by factorising $p(y\_{1:T})$ through time and replacing the integrals over latent variables by averages over particles. At time $t-1$ we have $K$ particles $k\_{t-1}=\lbrace z\_{1:t-1}^{(k)},g\_{1:t-1}^{(k)},m\_{1:t-1}^{(k)}\rbrace$ reflecting $K$ latent trajectories. Here $z\_t$ indicates the selected module at time $t$, $g\_t$ the corresponding hidden state of the gating RNN, and $m\_t$ the resulting hidden state of the module RNN. For each particle, we sample from the transition probability to obtain a predictive state $\lbrace z\_t^{(k)},g\_t^{(k)},m\_t^{(k)}\rbrace$. The likelihood of the observation $y\_t$ at time $t$ for this particle $k$ is given by the emission probability (Equation5):
> > \begin{equation*}
> > l\_t^{(k)}=p(y\_t|z\_t^{(k)}; \Lambda) = p(y\_t|mu\_t^{(k)};\Lambda)
> > \end{equation*}
> > where $\mu\_t^{(k)}=W\_M m\_t^{(k)}$ with $W\_M$ readout weights included in $\Lambda$. Ultimately at each timestep we want the predictive likelihood
> > \begin{equation*}
> > p(y\_t|y\_{1:(t-1)}; \Lambda) = \int p(y\_t|k\_t; \Lambda) p(k\_t|y\_{1:(t-1)}; \Lambda) dk\_t
> > \end{equation*}
> > The particle filter provides samples $k\_t$ which approximately follow $p(k\_t|y\_{1:(t-1)};\Lambda)$ so that we get a Monte Carlo approximation of the integral:
> > \begin{equation*}
> > p(y\_t|y\_{1:(t-1)}; \Lambda) \approx \frac{1}{K} \sum\_{k=1}^K p(y\_t | k\_t; \Lambda)
> > \end{equation*}
> > Therefore, using Equation 5, we get
> > \begin{equation*}
> > p(y\_t|y\_{1:(t-1)}; \Lambda) \approx \frac{1}{K} \sum\_{k=1}^K l\_t^{(k)}
> > \end{equation*}
> > which is Equation 7. The marginal likelihood of the whole sequence factorises through time:
> > \begin{equation*}
> > L = p(y\_{1:T}|\Lambda) = \prod\_{t=1}^T \frac{1}{K} \sum\_{k=1}^K p(y\_t|y\_{1:(t-1)}; \Lambda)
> > \end{equation*}
> > Substituting the particle estimate of each timestep's likelihood from Equation 7 yields
> > \begin{equation*}
> > L = p(y\_{1:T}|\Lambda) \approx \prod\_{t=1}^T \frac{1}{K} \sum\_{k=1}^K l\_t^{(k)}
> > \end{equation*}
> > to obtain Equation 8. In practise we calculate the log likelihood
> > \begin{equation*}
> > \text{log}(L) = \sum\_{t=1}^T \text{log}(\frac{1}{K} \sum\_{k=1}^K l\_t^{(k)})
> > \end{equation*}
> > and minimise $-\text{log}(L)$ by backpropagation through the particle filter.”_
> >
> > [CONTINUED BELOW]

---

> > > ### Author Response · Authors · 2025-11-20
> > > **Response (3/3)**
> > >
> > > > In addition, the objective function in learning is often either maximizing likelihood or minimizing loss. Eq. (8) is a likelihood and according to the paper: "is used as the training loss". Should it be the other way around, in which the negative log-likelihood is the training loss?
> > >
> > > Absolutely, thanks very much for catching this. The correct phrasing is that the negative log-likehood is the training loss, which corresponds to maximising the likelihood. We’ve removed “is used as the training loss” in line 153:
> > >
> > > _“We need that to calculate the marginal likelihood at each timestep
> > > \begin{equation}
> > > p(y\_t|y\_{1:t-1};\Lambda)=\frac{1}{K}\sum\_{i=1}^K l\_t^{(i)}
> > > \end{equation}
> > > so that we can get the marginal likelihood across the whole timeseries
> > > \begin{equation}
> > > L=p(y\_{1:T};\Lambda)=\prod\_{t=1}^T \frac{1}{K} \sum\_{i=1}^K l\_t^{(i)}
> > > \end{equation}
> > > We optimise model parameters $\Lambda=\lbrace \sigma, \theta, \phi, W\_G, W\_M\rbrace$ through gradient descent on negative log marginal likelihood $-\text{log}(L)$, backpropagating the loss through the particle filter on the training tasks.”_
> > >
> > > > Are the notations used in section 2,2 completely different from previous sections? Since $x\_t$ and $y\_t$ are used in previous sections, using them again with different meaning causes confusion.
> > >
> > > The notations are consistent, as in both cases $x\_t$ and $y\_t$ are the input and output at a particular timepoint in the timeseries. We hope this is clearer now given the explanation above, but would be happy to answer any follow-up questions.
> > >
> > > > The paragraph after Eq. (9) is confusing and requires further clarification. For example, why is suddenly a process of 6 shift operations introduced? Why is it 6?
> > >
> > > Thanks for bringing this up. Section 2.2 applies the architecture described in 2.1 to a synthetic set of tasks that we created specifically to highlight the model’s functionality. By using a hand-crafted (yet difficult!) task set, we make sure that we know the underlying true modularity and sequential structure. It allows us to carefully examine if the model recovers the ground truth. Here, our synthetic task set is defined by sequences of one of six possible shift operations on 6-dimensional vectors, where the 6 is entirely arbitrary. It’s just to show that our model can learn long sequences of arbitrary functions. This task delivers a proof-of-principle by demonstrating that the model recovers the shift modules and their sequencing rules that were used to generate task episodes. We’ve further clarified this in the manuscript:
> > >
> > > _“We first apply our model to an abstract rule learning task to show it can recover ground truth within-module and across-module dynamics. This task mimics the hardest variants of context-based rule switching because the model needs to simultaneously learn what the rules are and when to apply them, from uninformative input. We arbitrarily use vector dimension shifts as rules, with stay-switch transition dynamics, and random 6D vectors as inputs. There is nothing special about these choices; what matters is that they provide ground truth within-module and across-module dynamics that we can probe in the model. The data-generating process is given by …”_

---

### Meta-Review · Area_Chair_oPcy · 2026-01-09

**Summary:**

The authors propose to recast meta-learning as a probabilistic inference problem. They develop a RNN model, learning specific “subtasks” or motifs of a master task, acting as a task-solving backbone. In parallel, they implement a gating network that allows to guide information processing in the backbone model and select the appropriate RNN module sequence to solve a given task. At test time, their model can infer the best potential sequence of motifs to execute to solve a novel instance of a given task. They apply their model to (abstract) rule learning and motor learning.

Overall the reviewers had the following major concerns:
1. Tasks were too simplistic and toy.
2. Not enough baselines were compared to.
3. Paper was complicated and hard to understand.
4. Novelty and positioning to Hummos et al (2024) etc.
5. Assumption of temporal recombination may be limited.

**Reviewer Concerns:**

I found the overemphasis on comparison to Hummos et al 2024 (not currently published) as unnecessary. So I will discard that particular point of emphasis.

The major concerns that are not addressed are:

1. Tasks are too simplistic and toy.
2. Not enough baselines to compare to.
3. Paper is too complicated and hard to parse.

I think these three were debated but not really addressed. For instance, there was no comparison to Alet et al, which does modular meta-learning also. The idea of modular meta-learning is not so profoundly new, that it does not merit comparison. And in 2025, I think we need to see more than these 2 simplistic problems. Overall, I think those things make this paper not yet ready for publication.

**Reviewer Scores:**

rx9b would raise to 5/6 likely given clarifications.
6GAZ would stay the same at 8.
hQoD would likely raise a bit, but stay under 5 given the concerns about toy experiments and baselines were not resolved.
Gbhf seems a bit too stuck on a particular comparison to Hummos et al, so they wouldn't raise much probably. But I will discount that

Overall, the reviewers are reasonably fair (except this Hummos 2024 comparison) and I think the paper still stands at a reject.

---

### Decision · Program_Chairs · 2026-01-26

Reject